# The molecular basis for acetylhistidine synthesis by HisAT/NAT16

**Matti Myllykoski** [1] ✉, **Malin Lundekvam** [1], **Camilla Osberg** [1], **Solveig Siqveland Nilsen**[1] **& Thomas Arnesen** [1,2] ✉

Acetylhistidine has been detected in human blood, but its origin and function are not known. It is formed when the acetyl group of acetyl-CoA is transferred to the α-amino group of histidine. Here we identify the intracellular NAT16 as the human histidine acetyltransferase (HisAT) responsible for histidine acetylation in vitro and in vivo. A *NAT16* variant (p.Phe63Ser) present in over 5% of the population was previously found to correlate with reduced plasma levels of acetylhistidine and increased risk of kidney disease. Our biochemical analysis of HisAT/NAT16 Phe63Ser shows reduced affinity for Histidine supporting a model where this variant has less acetylhistidine catalysis leading to lower blood level of acetylhistidine. We find that HisAT adopts a double-GNAT (Gcn5-related N-Acetyltransferase) fold where the N-terminal domain binds acetyl-CoA and with distinct active site conformation allowing the binding of histidine in between the two domains. We detect similar structures from across living organisms and find that the HisAT structure is conserved in several archaeal and bacterial species. In sum, NAT16 is the human histidine acetyltransferase utilizing a rare double-GNAT structure to steer plasma acetylhistidine levels with potential impact for kidney function.

Histidine is an essential amino acid that is not synthesized in animals. It is incorporated into proteins and into abundant dipeptides such as carnosine (β-alanylhistidine), and it is decarboxylated to form the neurotransmitter histamine[1,2]. The imidazole side chain of histidine gives it unique properties among amino acids. Histidine can function as a pH buffer, proton donor or acceptor in enzymatic reactions, metal ion binder, and radical scavenger[2–4]. Blood histidine levels have been associated with disease states and health effects linked to e.g., cognition, inflammation, and glucose homeostasis[2,5]. The histidine side chain can be methylated and phosphorylated at both nitrogen atoms[6–8], and free histidine can be acetylated at the α-amine to form acetylhistidine (N-acetyl-L-histidine, NAH)[9].

Acetylhistidine is an abundant metabolite in ectothermic vertebrates. It has been detected from brain, lens and retina in the eye, heart, and skeletal muscle in several fish, amphibian, and reptile species[9–16]. Earlier reports suggested that acetylhistidine is largely absent from mammalian tissues, but it has been reported in some

studies from rat and guinea pig tissues and cells[15,17,18], and recent metabolomics analyses have found it to be present in human plasma[19–26] and cerebrospinal fluid[27]. The acetylhistidine concentration in fish lens can reach tens of μmol / g tissue. It has been suggested to function as an osmolyte or to participate in a molecular water pump, and its reduced abundance has been linked to cataract formation in farmed salmon[28–31]. However, the relevance of acetylhistidine in human blood is currently unknown. Acetylhistidine is synthesized from histidine and acetyl coenzyme A (Ac-CoA) by a histidine acetyltransferase (HisAT, EC:2.3.1.33). Fish HisAT was found to be encoded by the *NAT16* gene, but the corresponding human protein was found to be inactive towards histidine[32].

N-acylation is a covalent modification where an acyl chain is attached to an amine group. The substrates for N-acylation can be the N-termini or the lysine side chains of proteins or the amine groups of small molecules[33,34]. The most common type of N-acylation is N-acetylation, and, for example, 80% of human and plant proteins are

[1]Department of Biomedicine, University of Bergen, Bergen, Norway. [2]Department of Surgery, Haukeland University Hospital, Bergen, Norway. ✉e-mail: matti.myllykoski@uib.no; thomas.arnesen@uib.no

N-terminally acetylated with functional roles including the regulation of protein stability[35–42]. Lysine acetylation is a very common modification of histones and other proteins to regulate gene expression and other cellular functions[43–45]. The amino acid aspartate is acetylated at the α-amino group to form the important neurometabolite N-acetylaspartate by N-acetylaspartate synthetase NAT8L[46]. Many of the enzymes that catalyze N-acyltransferase reactions adopt a Gcn5-Related N-Acetyltransferase (GNAT) fold. The GNAT fold superfamily enzymes are found in all domains of life and catalyze the modifications of diverse substrates[47–50]. The enzymes in this superfamily tend to have low sequence similarity, reflecting the variable substrate pool, but they share the common structural features and the binding mode of the acyl-CoA co-substrate[51]. The GNAT fold itself is characterized by a central β-sheet typically composed of six or seven β-strands. The conserved acyl-CoA-binding site with the sequence Gln/Arg-x-x-Gly-x-Gly/Ala is located on one side of this sheet, and the substrate is bound on the other side. A V-shaped splay in the central sheet allows the acyl group to enter to the proximity of the primary amine of the substrate molecule[51]. In some rare cases the GNAT domain appears to have been duplicated but only one of the domains remains active. Examples of such double-GNAT structures include the eukaryotic N-myristoyltransferases (NMTs)[52] and bacterial clavulanic acid pathway acetyltransferase[53], mycothiol synthase[54], VipF acetyltransferase[55], and mycobacterial acetyltransferase Eis[56].

Here, we have studied the uncharacterized human NAT16 and found that this enzyme functions as a Histidine Acetyl Transferase (HisAT) both in vitro and in vivo. Our biochemical analysis of a HisAT/NAT16 genetic variant is consistent with a model where HisAT steers acetylhistidine plasma levels essential for normal kidney function. Crystal structures show that the human HisAT has a double-GNAT fold, and the analysis of the structural databases indicates that the HisAT subtype of the double-GNAT fold is found in all domains of life.

## Results

### Human NAT16 acetylates histidine in vitro

The protein encoded by the human *NAT16* gene remains functionally uncharacterized but is annotated as a likely acetyltransferase. To determine its biochemical function, the human NAT16 protein was expressed with a C-terminal 6xHis-tag in insect cells and purified using Ni$^{2+}$-affinity and size exclusion chromatography (Supplementary Fig. 1). The purified protein was screened for acetyltransferase activity using a 5,5′-dithio-bis(2-nitrobenzoic acid) (DTNB) assay in the

presence of Ac-CoA and potential small molecule and peptide substrates (Fig. 1a, Supplementary Table 1). Histidine, 1-methyl-histidine, and 3-methyl-histidine were the most active substrates (Fig. 1a, b). Some activity was seen also towards the positively charged amino acids arginine, lysine, and ornithine, and the bulky amino acids phenylalanine, tyrosine, and methionine. However, the reaction with histidine reached a similar CoA level in 2 min as with arginine in 30 min (Supplementary Fig. 1d). Other histidine derivatives including D-histidine, carnosine, and histamine did not react, but a low signal was obtained with histidinol (Supplementary Table 1). NAT16 did not acetylate selected peptides with different sequences and free N-terminal or lysine amino groups, including peptides with N-terminal histidine, suggesting that it is not an N-terminal acetyltransferase or a lysine acetyltransferase (Supplementary Table 1). The reaction was relatively stable at pH values of 7 and above and preferred a phosphate buffer (Supplementary Fig. 1e).

We determined the kinetic constants for NAT16 using a DTNB-based real-time assay (Table 1, Supplementary Fig. 2). The initial reaction rate was measured in the presence of different histidine and Ac-CoA concentrations (Supplementary Fig. 2a). The analysis of the results suggests that the acetyltransferase reaction proceeds via a tetrahedral intermediate, since when plotted on the graph [His]/$v_0$ vs. [His], the plots representing different Ac-CoA concentrations meet to the left of the [His]/$v_0$ axis (Supplementary Fig. 2b). The results also show that NAT16 suffers from considerable substrate inhibition at higher Ac-CoA concentrations (Supplementary Fig 2d, e). We also determined the apparent $k_{cat}$ and $K_M$ with arginine and lysine and found the $K_M$ to be an order of

### Table 1 | NAT16 kinetic constants for the mean of three independent series

|  | Mean | St. Dev. |
|---|---|---|
| $k_{cat}$ (s$^{-1}$) | 36.5 | 4.6 |
| $K_M^{Ac\text{-}CoA}$ (µM) | 19.8 | 4.5 |
| $k_{cat}$ / $K_M^{Ac\text{-}CoA}$ (s$^{-1}$ µM$^{-1}$) | 1.90 | 0.27 |
| $K_M^{His}$ (µM) | 215 | 53 |
| $k_{cat}$ / $K_M^{His}$ (s$^{-1}$ µM$^{-1}$) | 0.177 | 0.029 |

The assays measured $v_0$ at variable histidine concentrations (between 20 µM and 10 mM) with different fixed Ac-CoA concentrations (between 5 µM and 500 µM). Secondary plots were used to determine the kinetic constants. Source data are provided as a Source Data file.

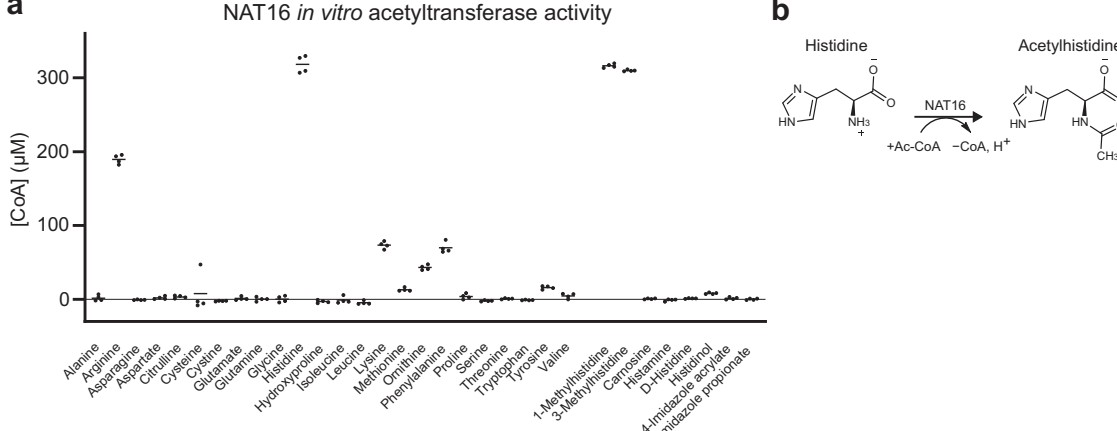

**Fig. 1 | Purified NAT16 acetylates histidine in vitro. a** Purified human NAT16 was studied with a DTNB assay in the presence of Ac-CoA and substrate candidates. The molecules that produced clear positive signals are shown in addition to selected negative hits. The Ac-CoA concentration was 300 µM, indicating that the histidine and both methylhistidine reactions reached the maximum during the 30-min reaction time. The assay consisted of four parallel technical replicate reactions and two control reactions without enzyme. Each substrate candidate was probed once. The individual data points and the mean for the parallels are shown. Source data are provided as a Source Data file. **b** Schematic reaction showing the NAT16 catalyzed conversion of histidine and Ac-CoA to acetylhistidine.

magnitude higher and $k_{cat}$ around 50-100 times lower than for histidine (Supplementary Fig. 2h, i).

## Human NAT16 is a cytosolic HisAT enzyme catalyzing cellular acetylhistidine formation

To study the function of NAT16 in human cells, we used the following set of cell lines: Human HAP1 wildtype (WT), HAP1 *NAT16* KO (KO), HAP1 wildtype stably overexpressing NAT16-V5 (WT + NAT16), and HAP1 *NAT16* KO stably overexpressing NAT16-V5 (KO + NAT16) (Fig. 2a, Supplementary Fig. 3a, b). Cell material from WT and KO cells overexpressing NAT16 was fractionated to different subcellular compartments. In both cell lines, NAT16 was found to co-localize with the GAPDH marker protein indicating that NAT16 is a cytosolic protein (Fig. 2b and Supplementary Fig. 3c).

Since our in vitro screen suggested that NAT16 acetylates histidine and perhaps other amino acids, these four cell lines were analyzed for changes in their metabolite levels. Profiling of polar and semi-polar metabolites from the cell pellets with liquid chromatography-mass spectrometry detected 816 compounds of which 132 were identified at different levels of certainty (levels 1, 2a, and 2b, see methods), and identity for further 138 compounds was proposed based on mass and elemental composition. The metabolites were extracted using a mixture of methanol and chloroform, and as a result, highly hydrophilic compounds would be excluded. Out of all the quantified acetylated molecules, only acetylhistidine was significantly decreased in the KO cells compared to the WT cells (Fig. 2c, Supplementary Data 1). NAT16 overexpression also strongly increased the acetylhistidine level in both WT and KO cells. Differences in the acetylhistidine levels between WT + NAT16 cells and KO + NAT16 cells can be attributed to differences in the overexpression levels between these cell lines (Supplementary Fig. 3b). The histidine levels were not affected by NAT16 (Fig. 2c). While the levels of N-acetylarginine and acetyllysine (the assay did not

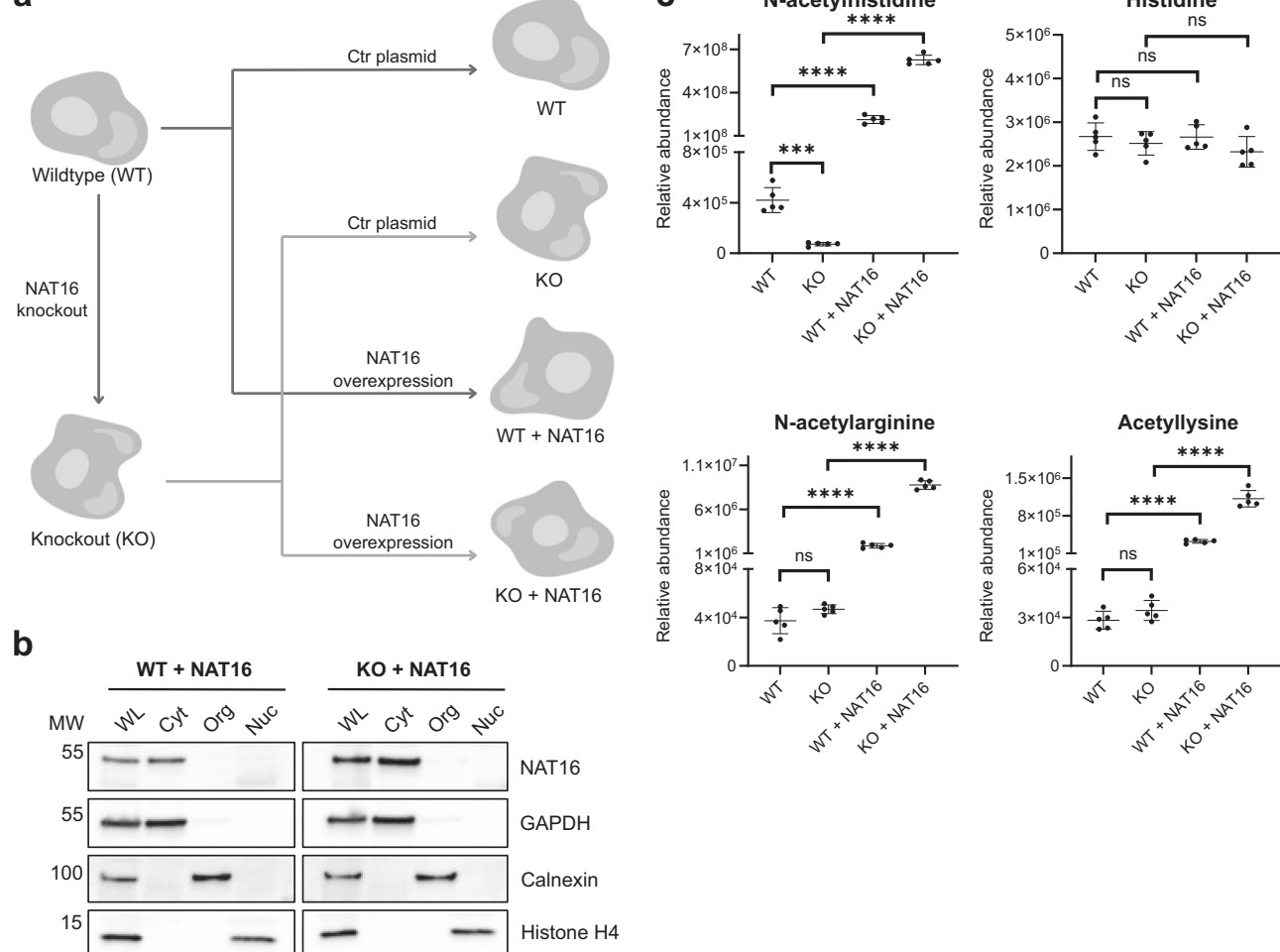

**Fig. 2 | NAT16 is responsible for acetylhistidine synthesis in human cells. a** A schematic showing the generation of HAP1 NAT16 knockout and overexpressing cell lines. Canva was used to generate the figure. **b** Cell fractionation of HAP1 WT cells and HAP1 NAT16 KO cells, both with NAT16-V5 overexpression to determine the cellular localization of the protein. Fractions of whole lysate (WL), cytosolic fraction (Cyt), organellar fraction (Org) and the nuclear fraction (Nuc) were analyzed by Western blotting. Different antibodies were included as controls for the different subcellular compartments: anti-GAPDH as a cytosolic control, anti-Calnexin an organellar control and anti-Histone H4 a nuclear control. Anti-V5 was used to detect NAT16-V5. The data are representative of three independent experiments. **c** Plots showing the relative abundance of selected metabolites detected in HAP1 cells with normal levels of NAT16 (WT), no NAT16 (KO), or overexpressed NAT16 (WT + NAT16; KO + NAT16). Each group had five replicates

and the individual points, their mean, and standard deviation are shown. Note that some of the Y-axes were split for clarity. The significance was estimated with a two-tailed *t* test coupled to Benjamini-Hochberg procedure with false positive rate at 0.05. Significance of the differences between the groups WT and KO, WT and WT + NAT16, and KO and KO + NAT16 are shown. The *p* values were as follows, Acetylhistidine: WT vs. KO: 0.000357, WT vs. WT + NAT16: 0.00000466, KO vs. KO + NAT16: 0.0000000147, Histidine: WT vs. KO: 0.534, WT vs. WT + NAT16: 0.985, KO vs. KO + NAT16: 0.390, N-acetylarginine: WT vs. KO: 0.157, WT vs. WT + NAT16: 0.00000852, KO vs. KO + NAT16: 0.0000000104, Acetyllysine: WT vs. KO: 0.219, WT vs. WT + NAT16: 0.00000989, KO vs. KO + NAT16: 0.00000669. The relative abundances are not comparable between different metabolites. Source data are provided as a Source Data file.

differentiate with certainty between α-Ac-Lys and ε-Ac-Lys) were not decreased in KO cells compared to WT cells, NAT16 overexpression increased the levels of both (Fig. 2c). This suggests that the endogenous NAT16 is not responsible for the acetylation of arginine and lysine, but NAT16 may acetylate these amino acids when the enzyme levels are very high. These cellular data fully agree with the in vitro biochemical data showing that human NAT16 is responsible for histidine N-acetylation, and we therefore name this protein Histidine Acetyl Transferase (HisAT).

## HisAT (NAT16) gene variant may affect blood acetylhistidine levels in the human population

After having established in vitro and in cells that NAT16 is the human HisAT, we wanted to further define its function. RNA expression profiles display the highest HisAT (*NAT16*) expression in the endocrine tissues such as hypothalamus, pituitary, and pancreas as well as in retina and the nervous system (https://www.proteinatlas.org/ENSG00000167011-NAT16/tissue) (Supplementary Fig. 4). This expression pattern could indicate that the acetylhistidine produced by HisAT-containing cells is secreted to blood by the endocrine tissues. Several metabolomics-coupled genome-wide association studies (GWAS) have suggested that specific variants in the *NAT16* locus were associated with variations in the acetylhistidine levels in serum and cerebrospinal fluid acetylhistidine levels[19–27]. One variant in particular, rs34985488-G, with a prevalence varying from 6.5% to 24.6% between different genetic similarity groups in the Genome Aggregation Database[57], was strongly associated with decreased serum acetylhistidine levels as compared to the more common variant rs34985488-A[20,21,24]. This variant is also associated with higher risk of diabetic kidney disease and chronic kidney disease[58]. The missense variant rs34985488-G introduces a phenylalanine to serine mutation at amino acid position 63 (F63S) of HisAT. (Fig. 3a).

To investigate whether HisAT F63S is affected in its capacity to acetylate histidine, we assessed the enzyme kinetics of HisAT WT and F63S (Fig. 3b, Supplementary Fig. 5a–c). The apparent kinetic parameters of WT and F63S HisAT were similar except for the 3.6-fold elevated apparent $K_M$ for histidine in F63S. This difference is expected to lead to slower acetylhistidine production, especially in low concentrations of histidine.

## HisAT has a double-GNAT structure

To further define the molecular basis for the substrate binding and specificity of HisAT, we crystallized it with different substrates and substrate analogs. The crystals of the full-length (369 aa) human NAT16 diffracted at best to around 2.6 Å. The N-terminus of human HisAT contains a region of around 50 residues preceding the predicted GNAT fold (Supplementary Fig. 6a). Since this region was predicted to be disordered, we deleted the residues 5-27 or 5-45 from the N-terminus of HisAT. The crystals of the truncated protein constructs diffracted to higher resolution and allowed us to solve the HisAT structure using single isomorphous replacement with anomalous scattering with three iodide-derivative datasets of the full-length construct (Table 2, Supplementary Table 2). The HisAT crystals usually adopted the space group *P*6₃ where the packing seemed to be dependent on the crystal contact formed by the second CoA moiety of a CoA disulfide molecule, formed by two CoA molecules covalently linked by a disulfide bond between their thiol groups, at the active site (Supplementary Fig 6b). The construct with the deletion 5-45 was also crystallized in another space group, *I*222, where it did not have the coenzyme A disulfide, but instead the truncated N-terminus interacted closely with another HisAT molecule (Supplementary Fig. 6c). In the *P*6₃ space group the first visible N-terminal residue was Glu47, while in the *I*222 space group the whole truncated N-terminus, residues 1–4 and from 46 onward, was visible. In both crystal settings the last visible residue was the first or second histidine of the C-terminal 6xHis tag and a single HisAT molecule was found in the asymmetric unit in both settings.

The overall HisAT structure is a double-GNAT fold composed of 10 α-helices and 13 β-strands (Fig. 4a, b). Together the β-strands 1–6 and 9-13 of the two GNAT domains form a continuous but distorted U-shaped β-sheet. The strands β6 and β13 are swapped between the two GNAT domains so that β6 is inserted between β12 and β13 and β13 is inserted between β5 and β6 (Fig. 4a). Most of the α-helical regions are outside this central sheet and only α1 and α2, and the corresponding helices from the second GNAT domain, α6, α7, and α8 are inside it. The substrate histidine binding site is formed by residues in the loop between α1 and α2 on one side and in the β-hairpin formed by β7 and β8 located in between α7 and α8 on the other side as well as the residues bordering the splay between β4 and β5. The conserved Ac-CoA binding residues are found in the β4-α3 loop and N-terminus of the α3 helix of the first GNAT domain and a β-bulge is formed by the residues Gly123 and Leu124 disrupting β4. The Ac-CoA interacting residues, and the β-bulge have been lost from the second GNAT domain (Supplementary Fig. 6d–f). Phe63 that influences the histidine substrate binding and the serum acetylhistidine levels (Ser63 of the rs34985488-G variant reduces serum acetylhistidine) is located in the α1 helix (Supplementary Fig. 6g).

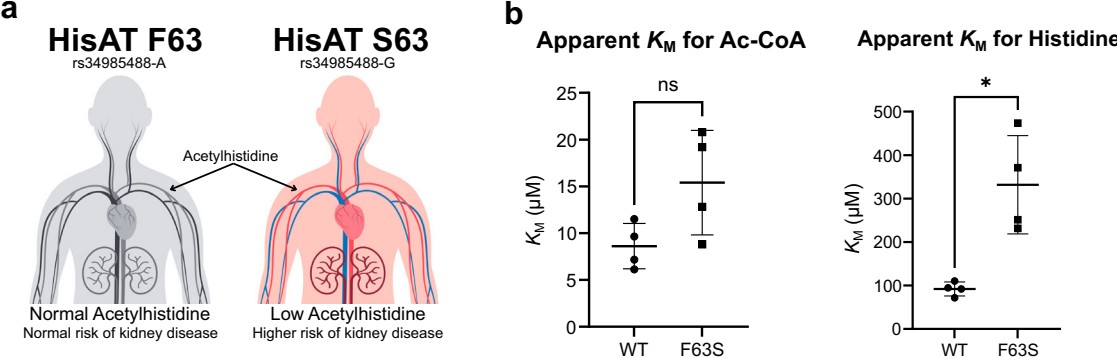

**Fig. 3 | The role of HisAT in controlling serum acetylhistidine levels. a** A model showing the relationship between the F63 (rs34985488-G) and S63 (rs34985488-A) variants and serum acetylhistidine levels. Canva was used to generate the figure. **b** Enzyme kinetics results show an increased apparent histidine $K_M$ for the HisAT S63 vs F63 variants. The assays measured $v_0$ at variable Ac-CoA concentrations (between 2 μM and 250 μM) at fixed histidine concentration (5 mM) and at variable histidine concentrations (between 20 μM and 4.5 mM) with fixed Ac-CoA (50 μM), both with four independent experiments. The individual $K_M$ values, their mean, and the standard deviation are shown. The two-tailed *p* values were determined with Welch's unequal variances *t* test. For Ac-CoA $K_M$, *p* value was 0.0885 (difference between means 6.79, 95% confidence interval −1.60 to 15.2) and for His $K_M$, *p* value was 0.0228 (difference between means 240, 95% confidence interval 62 to 417). Source data are provided as a Source Data file.

**Table 2 | Crystal data collection, data processing, and structure refinement statistics for merged reflections**

| | NAT16 d5-27_2 PDB: 9EMD | NAT16 d5-45_9 PDB: 9EMO | NAT16 d5-45_18 PDB: 9EMP | NAT16 d5-27_42 PDB: 9EMT | NAT16 d5-45_49 PDB: 9EN3 |
|---|---|---|---|---|---|
| **Data collection** | | | | | |
| Space group | $P\,6_3$ | $P\,6_3$ | $I\,2\,2\,2$ | $P\,6_3$ | $I\,2\,2\,2$ |
| Cell dimensions | | | | | |
| $a, b, c$ (Å) | 90.5 90.5 98.6 | 91.0 91.0 99.4 | 49.9 111.9 156.9 | 91.1 91.1 98.5 | 49.8 110.8 153.8 |
| $\alpha, \beta, \gamma$ (°) | 90 90 120 | 90 90 120 | 90 90 90 | 90 90 120 | 90 90 90 |
| Resolution (Å)* | 78.4–1.60 (1.70–1.60) | 78.8–1.90 (2.01–1.90) | 91.1-1.45 (1.54–1.45) | 41.8–1.40 (1.48–1.40) | 46.5-1.40 (1.48–1.40) |
| $R_{merge}$* | 0.060 (2.91) | 0.128 (3.89) | 0.052 (2.45) | 0.046 (2.96) | 0.053 (1.29) |
| $R_{meas}$* | 0.063 (3.02) | 0.135 (4.10) | 0.057 (2.67) | 0.048 (3.14) | 0.057 (1.40) |
| $CC_{1/2}$ (%)* | 100.0 (49.5) | 99.9 (32.3) | 100 (50.5) | 100 (32.6) | 99.9 (75.0) |
| $I\,/\,\sigma I$* | 21.90 (0.92) | 10.70 (0.55) | 16.41 (0.69) | 21.65 (0.61) | 15.6 (1.21) |
| Completeness (%)* | 99.9 (100.0) | 99.9 (99.8) | 96.6 (91.8) | 99.9 (99.5) | 99.2 (96.8) |
| Total reflections* | 846069 (135766) | 373,457 (58,412) | 503,876 (74,005) | 1,011,505 (131,447) | 610,914 (83,998) |
| Unique reflections* | 60,369 (9763) | 36,840 (5936) | 75,754 (11509) | 90,984 (14572) | 83,407 (13013) |
| Redundancy* | 14.0 (13.9) | 10.1 (9.8) | 6.7 (6.4) | 11.1 (9.0) | 7.3 (6.5) |
| Wilson $B$-factor (Å²) | 31.2 | 41.4 | 26.0 | 25.7 | 22.2 |
| $V_M$ (Å³/Da) | 2.9 | 3.2 | 2.9 | 3.0 | 2.8 |
| **Refinement** | | | | | |
| Resolution (Å) | 1.60 | 1.90 | 1.45 | 1.40 | 1.40 |
| No. reflections | 60,343 | 35,944 | 75,511 | 90,962 | 83,311 |
| $R_{work}\,/\,R_{free}$ | 0.167 / 0.185 | 0.223 / 0.239 | 0.188 / 0.209 | 0.160 / 0.180 | 0.166 / 0.183 |
| No. atoms | 3087 | 2802 | 3028 | 3225 | 3136 |
| Protein | 2606 | 2539 | 2629 | 2578 | 2658 |
| Ligand/ion | 139 | 114 | 86 | 222 | 81 |
| Water | 342 | 149 | 313 | 425 | 397 |
| Mean $B$-factors | | | | | |
| Overall (Å²) | 42.5 | 53.8 | 47.6 | 35.9 | 34.5 |
| Protein (Å²) | 40.7 | 53.3 | 46.6 | 32.5 | 32.7 |
| Ligands (Å²) | 61.3 | 73.9 | 79.6 | 59.6 | 64.5 |
| Water (Å²) | 48.7 | 48.5 | 47.2 | 44.2 | 40.0 |
| R.m.s. deviations | | | | | |
| Bond lengths (Å) | 0.006 | 0.007 | 0.009 | 0.004 | 0.008 |
| Bond angles (°) | 0.994 | 0.712 | 1.068 | 0.839 | 1.080 |
| Ramachandran | | | | | |
| Favored (%) | 99.4 | 99.1 | 98.8 | 99.1 | 99.1 |
| Allowed (%) | 0.6 | 0.9 | 1.2 | 0.9 | 0.9 |
| Outliers (%) | 0 | 0 | 0 | 0 | 0 |
| Rotamer outliers (%) | 0 | 0.4 | 0 | 0 | 0 |
| Clashscore | 0.9 | 2.1 | 0.7 | 1.1 | 0.7 |

*Values in parentheses are for highest-resolution shell.

The HisAT crystals with CoA disulfide were obtained in the presence of histidine, arginine, and imidazole. In addition, we obtained a crystal with CoA and N-myristoyl histidine and another with histidine and S-ethyl-CoA, which is an Ac-CoA analog used to mimic co-substrate binding (Supplementary Figs. 7, 8). We were unable to obtain crystals with Ac-CoA, Myr-CoA, or acetylhistidine, even when these molecules were present in the crystallization and crystal soaking solutions (Supplementary Table 3).

The substrate histidine side chain was positioned between the side chains of Tyr74 and Trp246 in a slightly staggered stacking position, and the imidazole group formed hydrogen bonds with two of the water molecules that surrounded the side chain (Fig. 4c). The histidine carboxyl group interacted with the hydroxyl groups of Tyr74 and Tyr79, the backbone amide of Gly123, and a water molecule. The histidine amino group interacted with the backbone carbonyl of Thr160

and another water molecule. In the structure with histidine and the CoA disulfide, an additional water was bound to the amino group, but in the structure with S-ethyl-CoA this water position was blocked by the ethyl group of the co-substrate analog (Supplementary Fig. 9a).

Arginine was the amino acid substrate with most activity in the substrate screening without an imidazole side chain (Fig. 1a). The crystal structure with arginine had the substrate residue occupying a very similar position to histidine (Fig. 4d, Supplementary Fig. 9b). The size of the side chain and the differences in hydrogen bonding disrupted the water chain that was present around the histidine side chain, so that some of the water molecules were displaced or not present (Supplementary Fig. 9b). The missing waters could also be explained by the comparatively poorer diffraction quality of this crystal (Table 2). The cause for the HisAT substrate preference of histidine above all other amino acids is not obvious from the structure

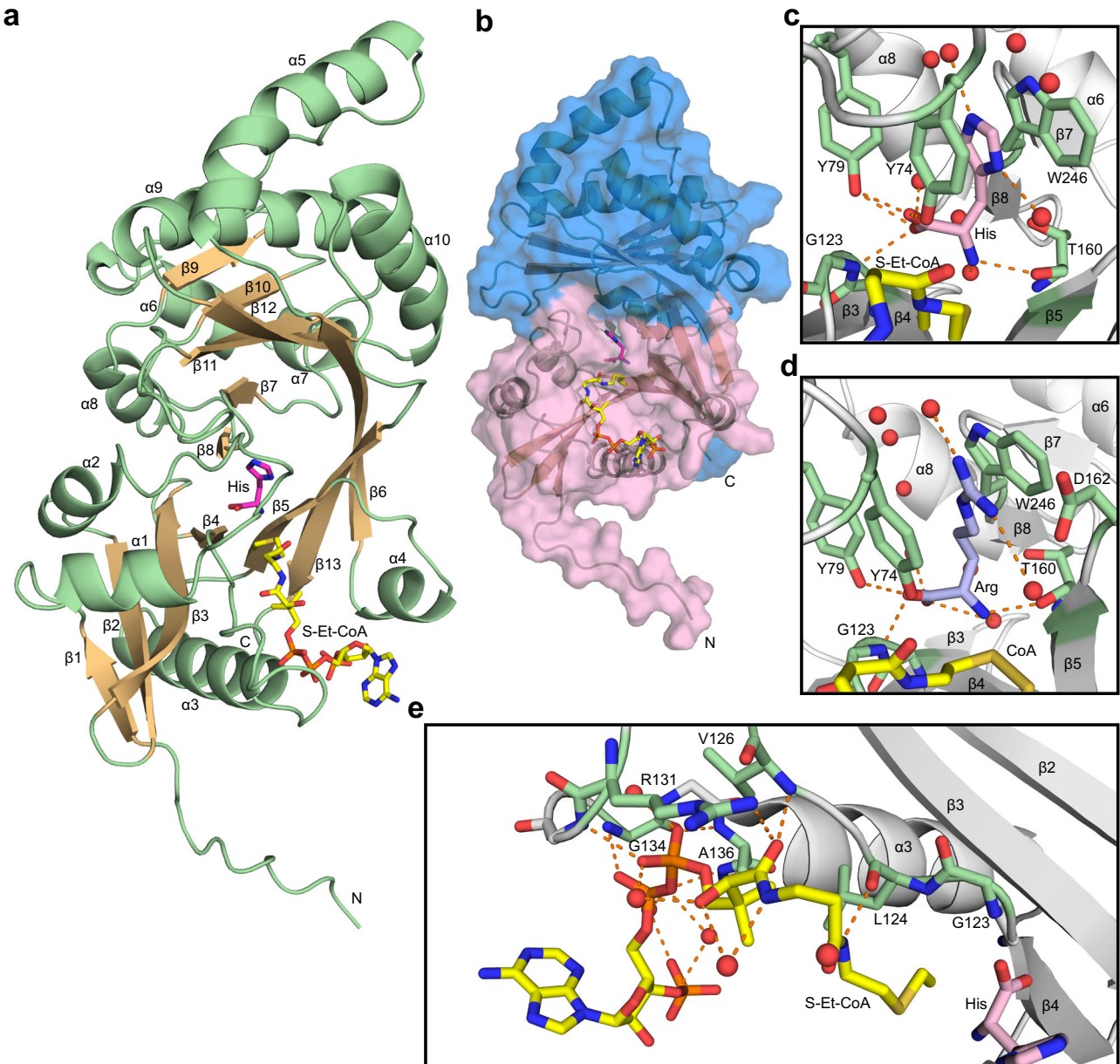

**Fig. 4 | HisAT crystal structure. a** HisAT adopts a double-GNAT fold composed of a central β-sheet and surrounding helices. The secondary structure elements are presented as a cartoon model where the α-helices and the loops are shown in green and the β-strands as light orange. The stick models of the substrate histidine (His) in magenta and the co-substrate analog S-ethyl-CoA (S-Et-CoA) in yellow are also shown. The N- and C-termini, the secondary structure elements, and the ligands are labeled. **b** A surface representation of the HisAT structure with the first GNAT domain (until res. 184) shown in pink and the second GNAT domain shown in blue. The stick representations of histidine and S-ethyl-CoA are shown in magenta and yellow, respectively. **c** Histidine shown in pink bound to the substrate-binding site, (**d**) arginine (Arg) shown in blue bound to the substrate-binding site, and **e** the Ac-CoA analog S-ethyl-CoA bound to the Ac-CoA binding site. The substrate and the co-substrate interacting residues are labeled and highlighted in green, and the surrounding waters are depicted as red spheres.

alone. In the structure where the buffer component imidazole was found in the active site, it was positioned in between Tyr74 and Trp246 in a similar way as the histidine side chain (Supplementary Fig. 9c, d). This affinity probably explains the difference in enzyme activity between histidine and arginine, but not the poor quality of the other aromatic side chain amino acids as HisAT substrates. A few acidic residues surround the substrate side chain and could promote via water interactions the binding of the basic histidine, arginine, and lysine side chains over phenylalanine and tyrosine (Supplementary Fig. 9e).

We used S-ethyl-CoA to mimic Ac-CoA binding because it likely adopts a position very similar to Ac-CoA. S-ethyl-CoA interacted in

canonical manner with the Arg131, Gly134, and Ala136 of the conserved GNAT domain acyl-CoA binding site. It also formed β-sheet-like interactions with Leu124 and Val126 (Fig. 4e). Based on the ethyl group position, the Ac-group oxygen of Ac-CoA would interact with the backbone amides of Gly123 and Leu124 that form the β-bulge. These two amides generate an oxyanion hole that stabilizes the ternary complex during the acetyl transfer. The crystals of the $P6_3$ space group invariably contained a CoA disulfide (Supplementary Fig. 9f). The first CoA of the CoA disulfide bound to the canonical position and aligned well with the other molecules bound in this position except for the 3′-AMP moiety that was poorly defined (Supplementary Fig. 8, Supplementary Fig. 9g). The second CoA of the CoA disulfide was positioned

in a cavity formed between the helices α3, α4, and the β-sheet segment composed of β5, β13, and β6.

## HisAT sequence and structure comparison

There are two structurally characterized double-GNAT fold proteins found in humans in addition to HisAT, the N-myristoyltransferases 1 and 2[59–61]. These enzymes transfer a myristoyl group from myristoyl-CoA to an N-terminal glycine or a nearby lysine in selected proteins[60,62–65]. We wanted to find out if HisAT and NMT share the NMT-type activity and if the two double-GNAT domains are homologous beyond the GNAT fold i.e., if the two proteins are the result of the same or separate GNAT domain fusion events.

To explore if HisAT and NMTs share functional similarity, we repeated the substrate screening assay with histidine and different types of acyl-CoA molecules (Supplementary Table 4). The activity was highest with the shorter acyl chains, but considerable activity was still present with myristoyl- and palmitoyl-CoA. We also tested the activity with HisAT in the presence of myristoyl-CoA and a selection of small molecules and peptides known or proposed to be myristoylated (Supplementary Table 5). The only clearly positive substrate was histidine. In order to understand the HisAT preference for acyl-CoA length, we determined its thermal stability using differential scanning fluorimetry (DSF) in the presence of CoA and different CoA derivatives. HisAT was slightly more stable in the presence of the longer chains, suggesting that it can interact with acyl chains longer than acetyl (Supplementary Table 6).

We also crystallized HisAT in the presence of Myr-CoA and histidine. The resulting structure incorporated a myristoylated histidine and a single CoA molecule (Fig. 5a, Supplementary Fig. 9g, h). We were unable to capture acetylhistidine in the crystal structures, and the presence of the myristoylated histidine here may indicate that the longer acyl chain gives this product higher affinity to the enzyme. The conformations of most of the residues in the active site are very similar between the structures with the histidine substrate and the myristoylated histidine product (Supplementary Fig. 9h). Some differences were still present. The substrate α-amide nitrogen was positioned closer to the Thr160 carbonyl group (2.69 Å) than the corresponding nitrogen in myristoyl histidine (3.06 Å). One of the waters coordinated by the imidazole was not present in the product structure. The interacting side chains of the nearby residues Arg161 and Asp163 were poorly defined in the electron density of the substrate structure in contrast to the myristoyl histidine structure and the structures with coenzyme A disulfide. The myristoyl chain occupied the same cavity between α3, α4, and the β-sheet segment composed of β5, β13, and β6 that was occupied by the second CoA moiety of the CoA disulfide (Supplementary Fig. 9g). This cavity is lined with several hydrophobic amino acids suggesting that the capacity to accept acyl-CoA co-substrates with longer acyl chains is a conserved feature for HisAT (Fig. 5b).

We compared the overall structures of HisAT and NMT1 and the position and environment of the myristoyl groups of myristoylated histidine bound to HisAT and myristoylated peptide substrate bound to NMT1. The structures show the similarity of the overall double-GNAT fold with notable differences in substrate binding and the active site (Fig. 5c). The active site of both enzymes is located at the interface of the two GNAT domains and the N-terminal domain binds the acyl-CoA and has the β-bulge that, for both enzymes, forms an oxyanion hole. In HisAT the only residue from the second GNAT domain that appears to contribute to enzymatic activity or substrate binding is Trp246. As NMT1 substrates are peptides or proteins with a specific N-terminal sequence, the second GNAT domain of NMT1 contributes more extensively to substrate binding. HisAT Trp246 is found in a β-hairpin formed by β7 and β8. A corresponding structure does not exist in NMT1 (Fig. 5d, e). In HisAT the protein C-terminus is located right after the last β-strand β13, while the NMT1 C-terminus is extended from

the corresponding strand, and the C-terminal residue Gln496 enters the active site and participates as the catalytic base in the enzymatic reaction (Fig. 5c–e)[65]. The myristoyl chains of myristoylated histidine in HisAT and myristoylated peptide product in NMT1 occupy very similar positions (Fig. 5d).

To explore the potential homology between HisAT and the NMTs, we first did a sequence similarity search of human protein sequences in the Refseq database. The search detected 11 different proteins and did not find the NMTs. The closest matches to HisAT were cysteine S-conjugate acetyltransferase NAT8, aspartate acetyltransferase NAT8L, and thialysine N-ε acetyltransferase SAT2. The detected proteins matched only the first GNAT domain. Multiple sequence alignment showed that there was little sequence conservation between these proteins beyond the Ac-CoA binding residues (Supplementary Fig. 10), which is a well-known feature of GNAT-fold proteins[47].

To study the HisAT sequence conservation, we searched the RefSeq database for similar sequences from different species. The search results indicated NAT16-like proteins are missing from several animals including mice and rats, birds, arthropods, and nematodes. Similar double-GNAT sequences were not detected in other multicellular eukaryotes and only from one unicellular eukaryote but were detected again in archaea (Supplementary Fig. 11). The alignment of the selected sequences showed strong conservation within vertebrates and in the first quarter of the sequence. The disordered N-terminus seemed to be largest in the mammalian sequences. Phe63, whose substitution to serine led to decreased plasma acetylhistidine levels, was well conserved among the selected animal sequences. Interestingly, some sequence features from vertebrates were better conserved in the archaeal sequences than in the non-vertebrate eukaryotes. The lack of *NAT16* gene in birds was previously reported[66]. In humans and those rodents that have it, *NAT16* is located between *MOGAT3* and *VGF*. In mouse and rat genomes this locus does not appear to contain *NAT16* and in mouse genome even the *Mogat3* gene is missing as described also previously (Supplementary Fig. 12)[67]. A great majority of non-primate mammalian *NAT16* sequences in the Refseq database are labeled as low quality protein, indicating that these sequences have been "modified relative to the genome sequence to correct for possible protein-altering mismatches or indels in the genome sequence"[68]. The reason for the abundance of such corrections here is not clear, but together with the lack of the gene in some rodent species it could indicate that *NAT16* has become a pseudogene in some mammalian lineages and is accumulating random mutations in the absence of purifying selection. To study this phenomenon further, we counted the number of *NAT16* and neighboring *MOGAT3* and *VGF* genes in the different mammalian orders (Supplementary Table 7). Both *NAT16* and *MOGAT3* were missing from many of the rodent species that had *VGF*. Only 23% of the *NAT16* sequences that were found were not labeled low quality protein, and most of these were primate sequences. Although many ungulate *MOGAT3* sequences were also labeled low quality, overall, this phenomenon was more abundant for *NAT16* than the other two proteins.

Since sequence similarity deprecates more rapidly than structural similarity[69], we searched for HisAT-like double-GNAT structures from PDB. The best hit was an unpublished entry of an archaeal double-GNAT-fold protein Ta0821 (PDB-ID 3C26)[70]. It was followed by many NMT structures, and the structure of a bacterial double-GNAT fold protein from the clavulanic acid biosynthesis pathway (PDB-ID 2WPX)[53,60]. HisAT and Ta0821 appear to be structurally more similar than HisAT and NMT1 (Fig. 5f), and they share the β-hairpin formed by β7 and β8 in HisAT that emerges from the second-GNAT domain (Fig. 5g). The tryptophan residue in this hairpin that participates in histidine substrate binding is also conserved in Ta0821 (Fig. 5g).

Based on the similarity of both sequence and structure, there are proteins homologous with HisAT in archaea. However, NMTs have

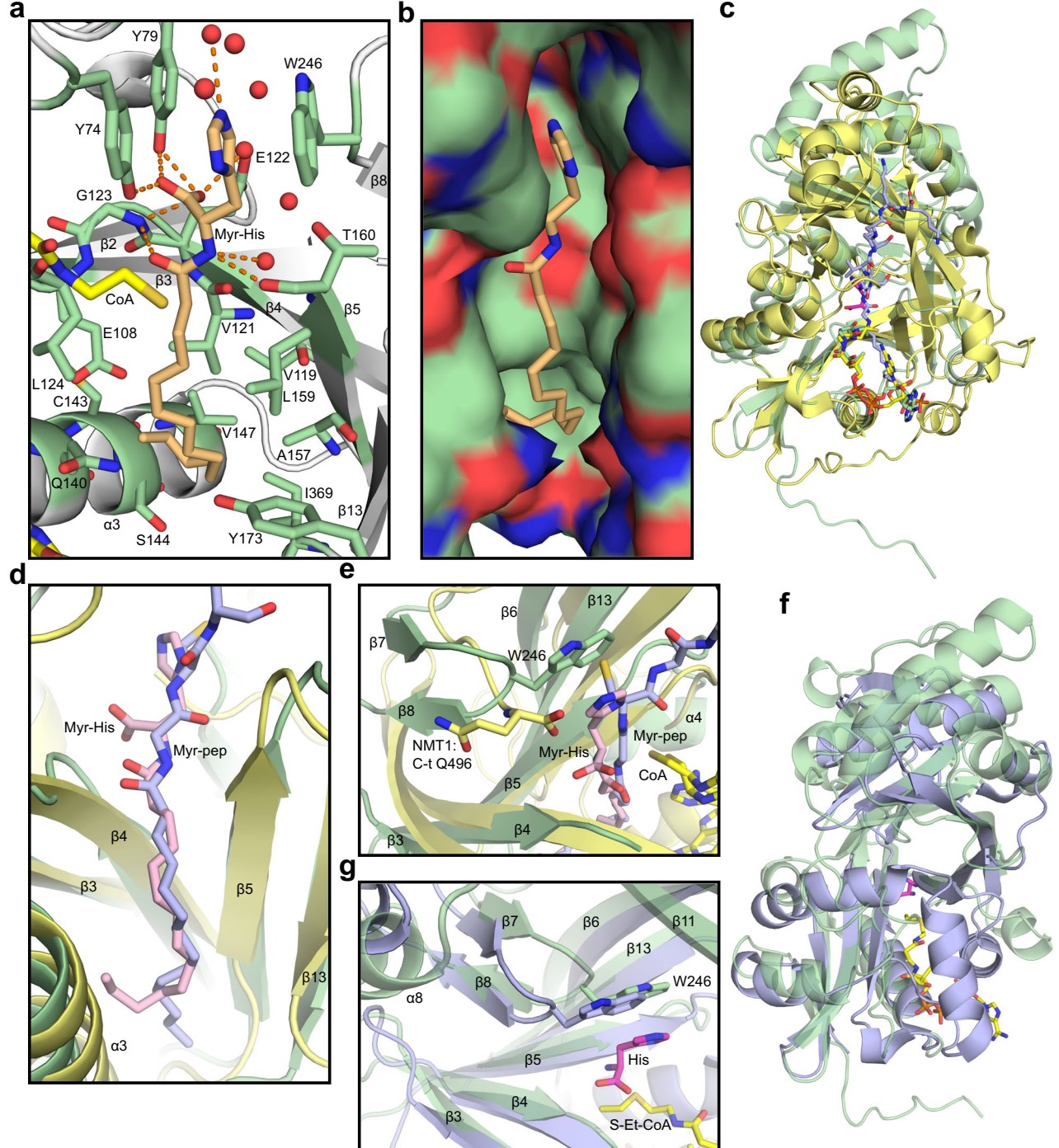

**Fig. 5 | Acyl chain binding and structural comparison between human HisAT and other double-GNAT enzymes. a** Binding mode of myristoylhistidine (Myr-His) and the residues that form a hydrophobic binding pocket identified in the HisAT crystal structure. **b** Surface model of the acyl chain binding pocket with oxygen and nitrogen atoms colored red and blue, respectively. **c** Overlay of the overall crystal structures of HisAT in green and human NMT1 (PDB-ID 5O9U)[65] in yellow. The alignment R.M.S.D of the Cα was 3.1 Å. **d** Overlay of the acyl chains of acylated peptide (Myr-pep) in NMT1 structure and acylated histidine (Myr-His) in HisAT structure. **e** Closeup of the overlay of NMT1 and HisAT active site structures showing the overlap of the catalytic C-terminal Gln496 of NMT1 and the histidine binding Trp246 of HisAT. **f** Overlay of HisAT and archaeal Ta0821 (PDB-ID 3C26)[70]. The alignment R.M.S.D. of the Cα was 2.6 Å. **g** Overlay of the β-hairpin structures at the active sites of HisAT and Ta0821.

been reported to be limited to eukaryotes[63]. To elaborate on the relationship of these protein groups, we searched PDB and Alphafold DB for structures of HisAT-like and NMT-like proteins with double-GNAT structures from animals, other eukaryotes, archaea, and bacteria. For the search we considered double-GNAT proteins with a similar β-hairpin as in HisAT formed by β7 and β8 as HisAT-type and proteins where the C-terminus ends up at the active site like in NMT1 structure as NMT-type (Fig. 5e). The HisAT-type structure search showed a similar pattern as the sequence search in that the HisAT-type structures were not found from plants or fungi but were found in some unicellular eukaryotes. The NMT-type structures were more abundant in eukaryotes. Surprisingly, in light of NMTs being reported to be

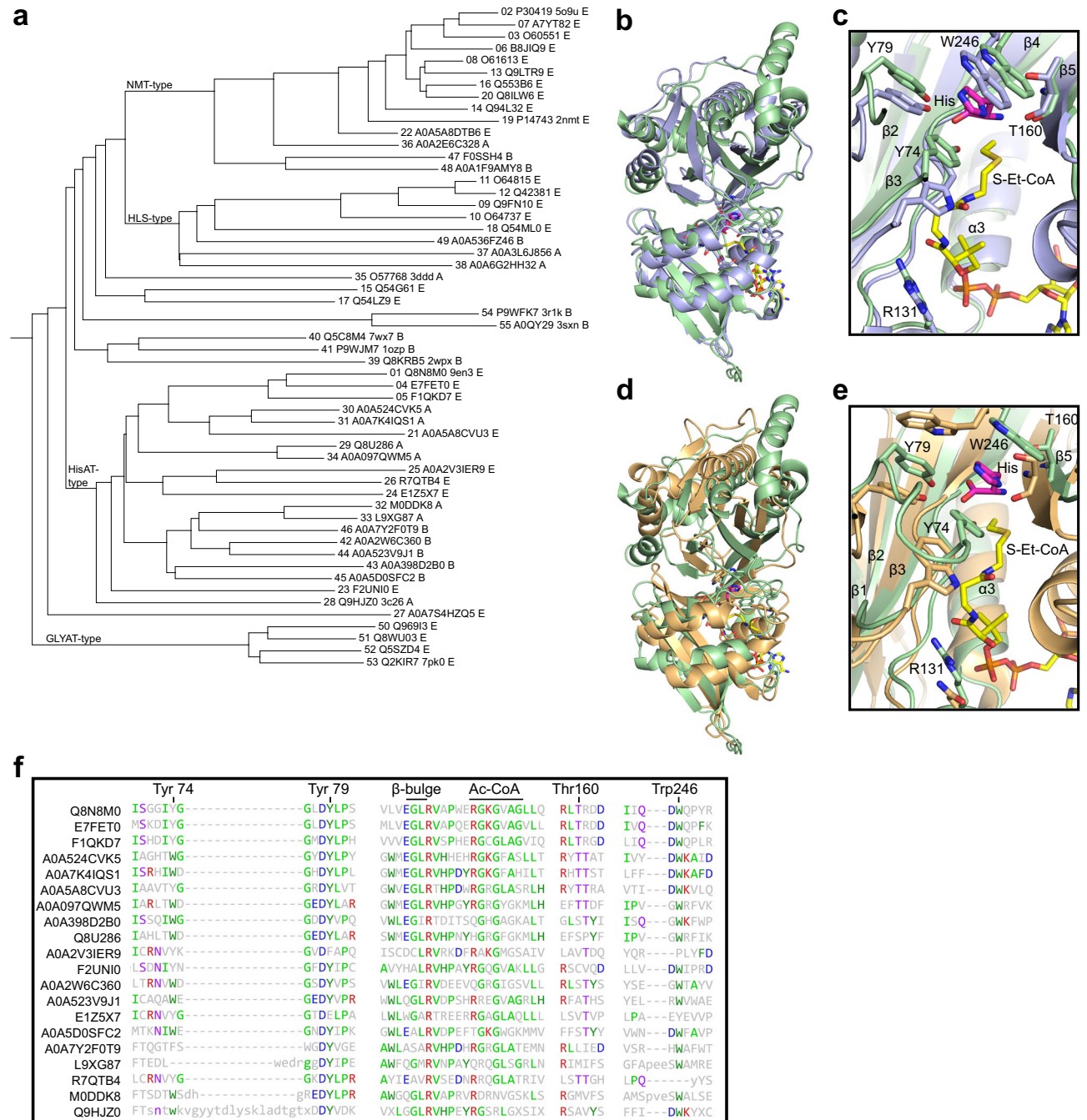

**Fig. 6 | Analysis of the structural relationships of HisAT-type and NMT-type double-GNAT proteins. a** Dendrogram of the structural relationships of selected protein structures determined by Dali server. The proteins are identified by a number included in Supplementary Table 9, Uniprot ID, possible PDB ID, and letter E for eukaryote, A for archaea, and B for bacteria. **b** Overlay (R.M.S.D. 2.4 Å) of the structures of human HisAT in green and a similar protein from *Thorarchaeota archaeon* (Uniprot ID A0A524CVK5) in blue. **c** Close-up of the overlay of the conserved active site residues of human HisAT and the archaeal protein. **d** Overlay (R.M.S.D. 3.8 Å) of the structures of human HisAT in green and a protein from *Chloroflexi bacterium* (Uniprot ID A0A2W6C360) in brown. **e** Close-up of the overlay of the conserved active site residues of human HisAT and the bacterial protein. **f** Structure-based sequence alignment of selected regions of the proteins in the HisAT-type group.

exclusively eukaryotic, both types of structures were detected in archaea and bacteria. The structure search also revealed that the mitochondrial glycine N-acyltransferases (GLYATs) have similar double-GNAT structures. We then did an all against all structural comparison of selected HisAT-type and NMT-type proteins, GLYATs, and some of the structurally characterized bacterial double-GNAT fold proteins that do not show clear similarity to either HisAT or NMT1 (Supplementary Table 8)[53–55].

The dendrogram derived from the all against all analysis of the structural relationships divided these double-GNAT proteins roughly into three groups (Fig. 6a). The first group consists of two subgroups identified by NMT-type proteins and by *Arabidopsis thaliana* Hookless (HLS). The previously characterized bacterial enzymes were assigned as early branches of this group. The HisAT-type structures form their own group where the two most distant branches were human HisAT and the archaeal Ta0841 (Fig. 6a). Interestingly, there were other

archaeal proteins structurally much closer to the mammalian HisAT enzymes such as a protein from *Thorarchaeota archaeon* (Fig. 6b, c). The HisAT-type group also included bacterial proteins (Fig. 6d, e). Several features of the HisAT-type active site, such as the histidine binding residues, the Ac-CoA binding residues, and the β-bulge, appeared to be conserved across the whole group (Fig. 6c, e, f). The mammalian GLYAT enzymes were assigned in a group of their own separated from the other structures (Fig. 6a). The dendrogram did not completely reflect the phylogenetic relationships, possibly because of the extreme evolutionary distances between the protein structures included in the analysis. However, the clear division to HisAT-type and NMT-type groups and the presence of both bacterial and archaeal proteins in both groups implies that both types were present in the last universal common ancestor or have experienced horizontal gene transfer between the domains of life after their divergence. Both alternatives make it difficult or impossible to determine if the two double-GNAT structure types originate from the same or different GNAT-fold fusion events.

## Discussion

The results presented here show conclusively that human *NAT16* encodes a HisAT. HisAT strongly acetylated histidine in vitro in the presence of Ac-CoA. It also acylated histidine in the presence of other acyl-CoAs, but these products accumulated more slowly. Acetylhistidine was detected in HAP1 cells even though they normally have little HisAT expression, and its level was lower in cells where *NAT16* was knocked out. Correspondingly, the acetylhistidine concentration increased massively when *NAT16* was overexpressed. The overexpression also increased the levels of acetylarginine and acetyllysine that were also produced by HisAT in vitro, but these molecules were not shifted in the knockout cells and, as they were also not acetylated to the same extent in vitro, are less likely to be important products in vivo. Acylhistidines with longer acyl chains were not identified in the metabolomics assay. Molecules with the same mass as propionyl- and butyrylhistidine were detected and were increased with NAT16 overexpression, but their identities were not confirmed. Acylated amino acids beyond acetylhistidine are produced in vivo and acylhistidines have also been reported occasionally from various organisms[71–74], but we are not aware of any indication that they are relevant for human physiology, and whether HisAT is a source of such molecules in vivo is not clear. Finally, genetic variation at the *NAT16* locus has been consistently linked to changes in plasma acetylhistidine levels[19–27]. One such variation produces a missense substitution of Phe63 with serine. HisAT with Ser63 was found to have a higher $K_M$ for histidine. The estimated HisAT catalytic rate at cellular histidine concentrations of 159 or 20 μM[75] would be 2- or 3-fold slower, respectively, with Ser63, potentially explaining the lower plasma acetylhistidine level.

HisAT structure is composed of two GNAT domains. The N-terminal domain binds Ac-CoA and the substrate histidine is bound in between the two domains. The substrate binding site combines the preference for aromatic and positively charged amino acid side chains, making histidine the preferred substrate, although we also obtained a crystal structure with arginine in the substrate position. Our results indicate that HisAT catalysis occurs via a ternary complex, and the ternary complex is likely stabilized by the oxyanion hole generated by the residues Gly123 and Leu124 of the β-bulge. GNAT-family enzymes commonly use the ternary complex mechanism, although substituted enzyme mechanism was also reported[51,76,77]. Similar double-GNAT structures have been described for NMTs and bacterial enzymes with varying substrates[53–55,65,78]. Typically, only one of the two GNAT domains is catalytically active. In HisAT, NMTs, and the mycobacterial Eis the N-terminal domain is active, while in several other bacterial double-GNAT enzymes the C-terminal domain is the active one[53–56,78]. In some cases, both domains may be able to bind Ac-CoA, and then the

presence of a β-bulge may be used to identify the active domain[53,54]. The β-bulge is not universal among functional GNAT domains, but it is present in most of them[34]. It is formed by one amino acid residue located in the last strand of the β-sheet before the Ac-CoA binding site splay turning its backbone 180° thus breaking the β-sheet. The atoms pointing toward the Ac-CoA and substrate binding sites may be the backbone carbonyl oxygens or, like in HisAT and NMT, the amine nitrogens forming the oxyanion hole.

In all double-GNAT enzymes the second GNAT is inserted in between the β6 and β7 strands of the canonical GNAT domain, suggesting a shared origin. In addition, both HisAT and NMTs are able to bind longer acyl chains such as myristate. We used conserved structural features to track structures similar to HisAT and the NMTs to find out if their common origin could be identified. Unfortunately, both types of folds were found in all three domains of life. This result suggests that these proteins either were present in the last universal common ancestor, or emerged in either archaea or bacteria and were transported to the other domain by horizontal gene transfer[79]. We should note that the presence of proteins that adopt the HisAT-type and the NMT-type structures in distantly related organisms should not be taken as proof that they share these enzymatic activities, regardless of the apparent similarity of the active sites. Minor changes in protein sequence may render them inactive or change their specificity and any presumed activity based on similarities in sequence or structure should be verified experimentally. On the other hand, histidine and Ac-CoA are ancient molecules, and it is not that farfetched that an enzyme with HisAT activity also emerged early.

Acetylhistidine is a human metabolite that we found to be present in cells even with relatively low amount of HisAT. The role of this molecule is not clear. Most considerations of its function have been focused on its high concentration in the eye lens of ectothermic vertebrates. Fish lens contains up to tens of mM of acetylhistidine and is highly dehydrated compared to its surroundings[11]. Baslow proposed in the 1960s that acetylhistidine enables a molecular water pump where it is released from the lens to the ocular fluid along its concentration gradient carrying along several water molecules against their concentration gradient. Acetylhistidine would then be broken down to histidine and acetate which would then be transported back into the lens without the water molecules and used to synthesize acetylhistidine[11,28]. In farmed Atlantic salmon, the amount of acetylhistidine in the lens was found to be linked to dietary histidine, and a lower amount of lens acetylhistidine increased the risk and the severity of cataracts in these fish[30,80]. A closely related salmonid Rainbow trout was less susceptible to cataracts with higher histidine and acetylhistidine levels in lens when reared in equal conditions[81]. Acetylhistidine has been proposed to be absent from mammalian lens and mammals altogether, although most of the studies looking into this have used mice and rats[10,12]. Our search for mammalian HisAT sequences in the Refseq database suggested that HisAT was missing from most rodents including mice and rats and could be a pseudogene in most other mammalian species but seemed to be intact in primates and few additional mammalian orders. These results would agree with the results that the human HisAT is an active enzyme that is expressed in several tissues and with the reports that acetylhistidine is detected in human blood and its levels are modulated by variations in the *NAT16* gene locus[19–27]. Horizontal cells in the retina were the site of the highest HisAT expression within human tissues and cells. It was also expressed in the brain and several secretory organs including the pancreas and the pituitary gland. Plasma acetylhistidine could originate from these secretory tissues. Lens cells were not included in the dataset, but metabolomic analyses of human lens have not reported the presence of acetylhistidine[82,83]. The potential pseudogenization of HisAT in other mammals needs to be independently confirmed. Another thing to consider is that acetylhistidine has occasionally been detected from tissues and cells of species that do not have *NAT16*, such as rats and

guinea pigs, suggesting either the chemical synthesis of acetylhistidine or the presence of another enzyme capable of histidine acetylation[15,17,18]. Another acetyltransferase NAT8 has been associated with the levels of several acetylated amino acids, including histidine, in metabolome studies[23,26], but the characterization of its activity in cellular lysates suggested it was only active towards cysteine-S-conjugates[84].

The function of acetylhistidine in human tissues is currently unknown, but there are some clues that could be followed. Acetylhistidine is structurally very similar to histidine and histidine dipeptides such as carnosine, anserine, and homocarnosine. These molecules have been proposed to have several properties emerging from the presence of the imidazole side chain such as pH buffering, metal binding, and radical scavenging[2,3,85–89]. Acetylhistidine could be expected to share these properties. However, acetylhistidine was suggested to be unable to prevent sugar-mediated protein cross-linking, in contrast to histidine and carnosine[90], so it is not clear to what extent these molecules are analogous. These histidine-containing molecules seem to share connections to glucose metabolism, kidney function, and diabetes. *NAT16* downregulation was linked to high glucose uptake in gastric cancer patient xenograft tissue[91]. *NAT16* was upregulated in renal biopsy samples of idiopathic nodular glomerulosclerosis[92]. Interestingly, the *NAT16* SNP (rs34985488-G) that leads to the F63S missense variant is associated with lower level of blood acetylhistidine. This allele is highly abundant in the human population reaching up to 25% in some genetic backgrounds, and it was recently reported to be associated with higher risk of diabetic kidney disease in type 1 diabetes and chronic kidney disease in a Finngen chronic kidney disease dataset[58]. Our biochemical data (Fig. 3b, Supplementary Fig. 5) suggests that the NAT16 F63S variant is impaired in acetylhistidine production and leads to lower blood levels. Since there is low expression of NAT16 in kidney, and high expression in endocrine tissues such as pancreas, it is likely that the increased risk of kidney disease observed for *NAT16* (rs34985488-G) is caused by diminished acetylhistidine blood levels, not local acetylhistidine levels in kidney tissue or cells. Histidine, carnosine, and the histidine degradation product imidazole propionate have also been associated with diabetes, glucose regulation, or kidney health[89,93–97]. In addition, lower serum carnosinase has been proposed to be associated with lower incidence of diabetic kidney disease[98,99]. A carnosinase homolog in fish called anserinase was found to break down acetylhistidine[100,101]. It could be interesting to determine if the plasma acetylhistidine levels are influenced by the serum carnosinase activity. Acetylhistidine could also serve as a pool of histidine unavailable for protein or dipeptide synthesis, or as proposed early on, a reservoir of acetyl groups[9]. The acetylhistidine molecular water pump mechanism was proposed specifically in the context of millimolar acetylhistidine present in the fish lens[11]. The acetylhistidine concentrations in human tissues are likely to be lower than that, and the presence of such system in humans is unclear. The highest level of HisAT mRNA expression was found in horizontal cells. The horizontal cells are retinal interneurons that modulate the output of the photoreceptor cells[102]. Intriguingly, the horizontal cell contributions to the photoreceptor synapse have been proposed to be mediated by a 'missing neurotransmitter' that possibly modulates the pH in the synaptic cleft[103]. As acetylhistidine is likely present in the horizontal cells following the HisAT expression there, can function as a buffering agent due to the presence of the imidazole side chain, and is likely to be secreted as it has been detected in plasma, perhaps it has some role in this horizontal cell function. One of the effects of excessive histidine supplementation was reported to be eye pain and difficulty focusing[104].

In summary, we have studied the function and structure of a previously uncharacterized human protein, NAT16. Our results firmly establish that this is the human histidine acetyltransferase, HisAT. This conclusion is also supported by several metabolomics-GWAS studies

that have linked *NAT16* variants to changes in plasma or cerebrospinal fluid acetylhistidine level. Our data supports a model where HisAT controls blood levels of acetylhistidine which determines the risk of kidney disease[58]. We also solved several crystal structures of HisAT with substrate histidine establishing the structural determinants for histidine acetylation.

## Methods

### Cloning, mutations, expression, and purification

Human NAT16 plasmid for protein purification was generated using GeneArt synthesis (Thermo Scientific) with a C-terminal 6xHis tag, inserted within pFastBac1 plasmid and codon-optimized for insect cell expression. To generate the deletion and missense mutations, the NAT16 pFastBac1 plasmid was linearized with PCR using primers (d5-27 forward: TCCTCCGAGACTCGT, d5-45 forward: CCTGAGGCTGAAGCT, d5-27 / d5-45 reverse: TTCCAACTTCATGGTGG, F63S forward: GAGCGCGAGTCCGAGGAAGTG, and F63S reverse: AGTAGCCACCAC-GAAGTCCAGG) that introduced the mutations, phosphorylated with polynucleotide kinase, and ligated with T4 DNA ligase, while the parental plasmid was digested with *Dpn*I. NAT16 plasmids were transformed into EMBacY *Escherichia coli* cells to generate the corresponding bacmids[105]. The purified bacmids were transfected to Sf9 cells (Thermo Scientific, 11496015) cultured in InsectXpress serum-free media (Lonza) using Baculofectin II transfection reagent (Oxford Expression Technologies Ltd). For expression, a 1:500 volume of amplified baculovirus was used to infect Sf9 cultures at 1 million cells / ml. The cultures were maintained at 1 million cells / ml, and the expression was continued for 72 h after the proliferation arrest. The cells were harvested by centrifugation, washed with PBS, and stored at −80 °C until purification.

The Sf9 cells that expressed NAT16 protein constructs were resuspended in lysis buffer containing 50 mM Tris-HCl pH 7.5, 0.3 M NaCl, 10% glycerol, 20 mM imidazole, 0.1% Triton X-100, and 1x EDTA-free protease inhibitor cocktail (PIC) (Roche), lysed with glass Dounce homogenizer, and centrifuged to remove insoluble material. NAT16 was purified from the soluble lysate with IMAC using either His-trap HP column (Cytiva Life sciences), or His-pur Ni-NTA matrix (Thermo Scientific). Column or matrix was washed with lysis buffer without Triton and protease inhibitor, and bound proteins were eluted with the same buffer but containing 0.3 M imidazole. NAT16 protein constructs were further purified with size-exclusion chromatography using Superdex 75 column with a running buffer containing 10 mM Tris-HCl pH 7.5, 150 mM NaCl, and 10% glycerol. Purified NAT16 was concentrated, aliquoted, flash-frozen in liquid nitrogen and stored at −80 °C until use. Some NAT16 activity was apparently lost when the enzyme was stored in this manner for several months.

### Substrate screening

Substrates for NAT16 were screened using the DTNB method described by Foyn and colleagues[106]. Briefly, substrate candidates were incubated in 50 μl volumes with 300 μM Ac-CoA and 500 nM NAT16 in reaction buffer containing 50 mM Na-phosphate pH 7.5, 100 mM NaCl and 1 mM EDTA. Substrate concentration was 300 μM for peptides and 2 mM for other molecules. Reactions were incubated for 30 min at 37 °C and quenched with Gd-HCl containing buffer. DTNB was added to around 3 mM and the 2-nitro-5-thiobenzoate absorbance was measured at 412 nm from 96-well plates using Tecan Infinite M Nano plate reader (Tecan). The plate pathlength was determined and the amount of CoA formed was calculated from the absorbance after pathlength correction using the extinction coefficient 13700 $M^{-1}$ $cm^{-1}$[106,107]. For the histidine time curve, the reactions prepared as described above were stopped at different time points. For the pH and buffer screening, the reaction buffer was changed to 50 mM buffer as indicated in Supplementary Fig. 1e, with the enzyme concentration as 100 nM, and the reaction time as 10 min.

## Enzyme kinetics assays

Enzyme kinetic data were collected using an adapted DTNB assay. The data for the wild type NAT16 were collected in three separate series using three separate expressions and purifications of the enzyme. Ac-CoA concentration ranged from 5 to 500 μM and L-His concentrations were 0, 20, 50, 120, 300, 720, 1800, or 4500 μM for the three lower Ac-CoA concentrations in Series 1 and 2 and for all Ac-CoA concentrations in Series 3, and 0, 50, 120, 300, 720, 1800, 4500, or 10,000 μM for the three higher Ac-CoA concentrations in Series 1 and 2. The reactions contained 50 mM Na-phosphate pH 7.5, 100 mM NaCl, 1 mM EDTA, 0.02% Tween-20, 1 mg/ml BSA, and 100 μM DTNB. NAT16 was 20 nM. The reactions were pre-incubated at 37 °C on 96-well plates prior to the enzyme addition and kept at 37 °C for the duration of the measurement. Tecan SPARK 20 M (Tecan) at the core facility for Biophysics, Structural Biology, and Screening (BiSS) at the University of Bergen, or the Tecan Infinite M Nano plate reader (Tecan) was used to determine the absorbance at 412 nm. The well pathlength was determined by subtracting the absorbance at 900 nm from absorbance at 975 nm, both measured at the end of the reactions. The result was divided by 0.18 to obtain the pathlength in cm[107]. The CoA concentration was calculated from the 2-nitro-5-thiobenzoate absorbance at 412 nm using extinction coefficient 13800 $M^{-1}$ $cm^{-1}$[108]. The reaction progress curves were input to ICEKAT server, where the initial rates for each reaction were estimated using logarithmic approximation[109,110]. The initial rates were used to determine the apparent $K_M$ and $V_{max}$ values using least squares fitting of the Michaelis-Menten equation in Graphpad Prism (GraphPad Software, LLC) initially for each Ac-CoA concentration separately. Primary plots of [His]/$v_0$ vs. [His] were generated to study the enzyme mechanism. As these plots met left of the Y-axis, the reaction was assumed to follow the ternary complex mechanism and was modeled using the equation $v_0 = V_{max} \cdot$ [His] $\cdot$ [Ac-CoA] / ($K_i^{His} \cdot K_M^{Ac-CoA} + K_M^{AcCoA} \cdot$ [His] $+ K_M^{His} \cdot$ [Ac-CoA] $+$ [His] $\cdot$ [Ac-CoA])[111]. As the apparent $V_{max}$ ($V_{max}^{App}$) determined by varying histidine at different fixed Ac-CoA concentrations, can be described by the equation $V_{max}^{App} = (V_{max}$ [Ac-CoA]$) / (K_M^{Ac-CoA} +$ [Ac-CoA]$)$, which is of the form of the Michaelis-Menten equation, $V_{max}$ and $K_M^{Ac-CoA}$ could be determined from the double-reciprocal secondary plot of $1/V_{max}^{App}$ vs. $1/$[Ac-CoA] generated using the $V_{max}^{App}$ values derived from the fits of the Michaelis-Menten equation with variable histidine at different fixed Ac-CoA concentrations[111]. $V_{max}$ was derived from the inverse of the Y-axis intercept and $K_M^{Ac-CoA}$ from the negative inverse of the X-axis intercept. The $k_{cat}$ values were calculated from the $V_{max}$ values by dividing with the enzyme concentration. The ratio of the apparent constants $V_{max}^{App} / K_M^{App}$ can be represented in Michaelis-Menten form by the equation $V_{max}^{App} / K_M^{App} = ((V_{max} / K_M^{His})$ [Ac-CoA]$) / (((K_i^{His} K_M^{Ac-CoA}) / K_M^{His}) +$ [Ac-CoA]$)$[111]. Thus, another double-reciprocal secondary plot of $K_M^{App} / V_{max}^{App}$ vs. $1/$[Ac-CoA] could be used to determine $K_M^{His} / V_{max}$, from the Y-intercept, and $K_M^{His}$ was calculated from this value by multiplication with $V_{max}$ derived as described above. As Ac-CoA inhibited the enzymatic reaction at higher concentrations, only the apparent values from the lower Ac-CoA concentrations in Series 2 and 3 were used to estimate these values. The $k_{cat} / K_M$ values were calculated from the $k_{cat}$ and $K_M$ values.

To compare directly the WT enzyme and the F63S variant, the apparent values for the kinetic constants were determined while varying only one substrate at a time. The reactions were carried out and analyzed in the same way as above, but the histidine series (0, 20, 50, 120, 300, 720, 1800, or 4500 μM) were only done at a single Ac-CoA concentration of 50 μM and the corresponding Ac-CoA series (0, 2, 5, 10, 25, 50, 125, or 250 μM) only at a single histidine concentration of 5 mM. The variable histidine series comparing WT with E108Q and E122Q variants was done in the same way. To test the activity with arginine and lysine, we again did the assays with fixed Ac-CoA concentration of 50 μM and varied only arginine (0, 240, 600, 1440, 3600, 9000, 21600, or 60000 μM) or lysine (0, 600, 1440, 3600, 9000,

21600, 60000, or 150000 μM). With histidine as substrate, NAT16 concentration (WT or F63S mutant) was 20 nM and with arginine and lysine as substrate it was 50 nM.

## Differential scanning fluorimetry

To determine the stability of NAT16 in the presence of CoA and different acyl-CoA derivatives, the protein was subjected to DSF. NAT16 at 0.3 mg/ml was applied on three wells on a Lightcycler 480 Multiwell plate 384 (Roche) with a particular additive as listed in Supplementary Table 6. The buffer was the same as that used for substrate screening. CoA and acyl-CoA concentrations were 10 and 100 μM. SYPRO Orange protein gel stain (Thermo Fisher Scientific) was at 5X concentration. The plate was gradually heated from 10 to 95 °C while the fluorescence from the dye was measured using a CFX384 Touch Real-Time PCR detection system (Bio-Rad) at the BiSS core facility. The results were analyzed, and the melting temperatures were determined using the DSFworld website[112].

## Crystallization

The crystals of the different NAT16 constructs were grown using sitting drop vapor diffusion on TTP Labtech 96-well 3-drop plates. The drops were pipetted with Mosquito LCP (TTP Labtech) at the BiSS core facility. The crystallization and cryoprotection conditions are listed in Supplementary Table 3. Protein concentrations were between around 5 and 9 mg/ml. The crystallization plates were incubated at 4 °C or 8 °C. For iodide derivatization, sodium iodide was included in the cryoprotection solution and allowed to soak in for a few minutes. The crystals incubated with the cryo-solution were flash-frozen in liquid nitrogen and stored at a cryogenic temperature until measurement. The Ac-CoA mimic S-ethyl-CoA was purchased from Jena Bioscience.

## Crystal data collection and processing, structure solution, refinement, and analysis

Crystal data were collected at DESY beamline P11 at wavelength 1.033 Å or 2.066 Å (iodide datasets) and at 100 K using CrystalControl software or at EMBL beamline P14 at wavelength 0.976 Å and at 100 K using mxCuBE software at the Petra III storage ring in Hamburg, Germany[113]. The data were processed with XDS[114]. Data collection statistics are presented in Table 2 and Supplementary Table 2.

Single isomorphous replacement with anomalous scattering was used to solve the phases for the initial structure. Three datasets from two iodide-derivatized crystals of full-length NAT16 were scaled together with XSCALE[114], and the combined dataset was used as the derivative dataset. The dataset of the crystal NAT16d5-27_2 was used as the native dataset. SHELXC was used to prepare the data, SHELXD was used to determine the anomalous substructure, and SHELXE was used to trace the poly-alanine chain[115]. Phenix.autobuild[116] was used to build the HisAT structure. The subsequent structures were solved with molecular replacement using phenix.PHASER[117] and the refined NAT16d5-27_2 crystal structure as the template. All of the structures were refined with phenix.refine[116] and Coot[118]. The structures were validated using Molprobity[119] and the PDB validation system[120]. The ligand omit maps were generated using phenix.polder[121]. Structure refinement statistics are presented in Table 2.

For visualization and analysis, protein structures were aligned using Secondary Structure Matching algorithm[122]. Coot and PyMOL (Schrödinger, LLC) were used for structure analysis, and PyMOL was used to generate the structure figures. Structures similar to human HisAT and human NMT1 (PDB-ID 5O9U)[65] were searched from the PDB database[123] and AlphaFold Protein Structure Database[124] using Dali server[125] and Foldseek[126]. The selected structures (Supplementary Table 8) were compared using the all against all structure comparison on the Dali server. A separate comparison was done for the HisAT-type proteins to get a clear structure-based sequence alignment.

Archaeopteryx was used to plot the Dali dendrogram output[127]. The secondary structure disorder was predicted using DISOPRED3[128].

## Sequence similarity analysis

Delta-BLAST[129] of the Refseq database[130] was used to search for similar sequences to human NAT16. The selected sequences were aligned using MAFFT[131] and displayed using UGENE[132]. Genome organizations in the MOGAT3-NAT16-VGF locus for human, Eastern gray squirrel, mouse, and rat were downloaded from the NCBI Gene database[133]. For the analysis of low quality protein sequences, Delta-BLAST was used to search with the human NAT16, MOGAT3, and VGF sequences for mammalian sequences in the Refseq database. These were assigned to different orders according to Murphy et al.[134], and the number of sequences marked 'Low quality protein' or significantly truncated were counted.

## NAT16-V5 knock-in in HAP1 WT and NAT16 KO cells

HAP1 WT (Horizon Discovery, C631) and NAT16 KO cells (Horizon Discovery, HZGHC007792c003) were grown in Iscove's Modified Dulbecco's Medium (IMDM, biowest #L0191-500), 10 % Fetal Bovine Serum (FBS, Sigma-Aldrich #F7524), 1 % Penicillin-Streptomycin (Sigma-Aldrich #P0781) at 37 °C and 5 % $CO_2$. Both cell lines were subjected to NAT16-V5 knock-in using the PiggyBac transposon system (VectorBuilder), including a custom-made plasmid VB200923-1149dfh pLV[Exp]-Neo-CMV > {NAT16-spacer-V5(ns)}/V5:P2A:EGFP. Approximately 2 million cells were seeded in a 10 cm dish and transfected four hours later with 3 µg NAT16-V5-P2A-EGFP plasmid and 3 µg helper plasmid with mCherry (VectorBuilder, # VB210324-1057jnh) in 18 µl X-tremeGENE™ 9 (Roche, #6365779001), 500 µl Opti-MEM (Gibco, #31985070) mix. Forty-eight hours post transfection, the cells were prepped for single cell sorting into three 96 well plates (Sarstedt #83.3924.500) containing 200 µl 20 % FBS, 1 % Penicillin-Streptomycin, IMDM cell medium. They were trypsinated (Gibco, # 25300054), washed in 1x Phosphate buffered saline solution (PBS, Gibco, #18912014, 1xPBS), harvested at 300 x g for 5 min, and finally resuspended in roughly 3 ml IMDM. EGFP- and mCherry-containing cells were detected with filter FL1 (525/50) and FL3 (617/30), respectively and single sorted on a Sony SH800 cell sorter using Flowjo software at the Bergen Flow Cytometry Core Facility (Supplementary Fig. 13). Background signal was detected in non-transfected cells and cells transfected with either of the two plasmids to determine the gating of target cells. Cell sorting was done with the lowest flow pressure possible, aiming for a maximum of 500 cells per second. Clones were screened using pre-cleared cell lysates for SDS-PAGE and Western blot with anti-V5 (Invitrogen # R960CUS). Negative clones served as mock cell lines for further metabolomics analysis.

## Verification of NAT16 KO

Total RNA was isolated from WT and NAT16 KO cells using RNeasy mini kit (Qiagen #74104) and the quality was checked on a 1 % agarose gel. Subsequently, cDNA was synthesized from 500 ng RNA using iScript cDNA synthesis kit (BIO-RAD #170-8891) and used as template for the amplification of NAT16 and ACTB with Platinum SuperFi PCR master mix (Invitrogen #12358010). Water served as the negative control. The primers that were used were CATCCTTTTGGTCCGATTCAACG (NAT16, forward), CGCAGGTTGCTTTCGCTAGGCC (NAT16, reverse), CCGCCAGCTCACCATGGATGA (ACTB, forward), and CTA-GAAGCATTTGCGGTGGACGATGG (ACTB, reverse).

## Metabolomics analysis

All four PiggyBac HAP1 cell lines were harvested for metabolomics analysis. We seeded 2.5 million cells in five replicate 10 cm dishes of each cell line (Sarstedt #3902) and harvested them in low protein-binding microtubes (Sarstedt #72.706.600) after 24 h; washed the cells two times in the dish with ice-cold 1xPBS, and scraped them in ice-

cold 1xPBS before centrifugation at 300 x g for 8 min at 4 °C. The cell pellets were stored at -80 °C.

Extraction of cell samples and both polar and semi-polar LC-MS/MS analyses were performed at the MS-Omics company, part of Clinical Microbiomics (Vedbæk, Denmark). The cell pellets were extracted in 300 µl of a mixture of 4:1 (vol/vol) methanol and chloroform. After mixing and centrifugation for 5 min at 3000 x$g$ at 4 °C, the whole supernatant was split into two aliquots, evaporated under a gentle flow of $N_2$, and reconstituted in the same volume. The aliquot for polar analysis was resuspended in equal mixture of the A and B mobile phase eluents (HILIC). The aliquot for semi-polar analysis was resuspended in 10%B (Reverse phase). Stable isotope-labeled internal standards were added to all samples. Reconstituted extracts were filtered through a Spin-X centrifuge tube filter (0.22 µm, Corning Costar) using centrifugation at 15,000 x g for 5 min at 4 °C. To ensure high-quality sample preparation, a quality control (QC) sample was prepared by pooling small equal aliquots from each sample, to create a representative average of the entire set. This sample was treated and analyzed at regular intervals throughout the sequence.

The polar metabolite profiling analysis was carried out using a Vanquish LC (Thermo Scientific) coupled to a Q Exactive HF MS (Thermo Scientific). The UHPLC was performed using a InfinityLab Poroshell 120 HILIC-Z column (2.7 µm, 150 mm × 2.1 mm, Agilent). System calibration was performed at once a week using Pierce™ FlexMix™ Calibration Solution (Thermo Scientific) and included low-mass contaminants for extended mass range calibration. An electrospray ionization interface was used as the ionization source. Analysis was performed in negative and positive mode. The UPLC was performed using a slightly modified version of the protocol described previously[135] (Supplementary Table 9).

The semi-polar metabolite profiling was carried out using a Vanquish LC (Thermo Scientific) coupled to an Orbitrap Exploris 240 MS (Thermo Scientific). The UHPLC was performed using an adapted version of the protocol described previously[136] (Supplementary Table 9). An electrospray ionization interface was used as the ionization source. Analysis was performed in positive and negative ionization mode under polarity switching. System calibration was performed at once a week using Pierce™ FlexMix™ Calibration Solution (Thermo Scientific) and included low-mass contaminants for extended mass range calibration.

Metabolomics processing was performed untargeted using Compound Discoverer 3.3 (Thermo Scientific) and Skyline 22.2 (MacCross Lab Software) for peak picking and feature grouping, followed by a in-house annotation and curation pipeline written in MatLab (2022b, MathWorks). Identification of compounds were performed at four levels; Level 1: identification by retention times (compared against in-house authentic standards +/−0.2 min), accurate mass (with an accepted deviation of 3ppm), and MS/MS spectra (match factor threshold >60), Level 2a: identification by retention times (compared against in-house authentic standards), accurate mass (with an accepted deviation of 3ppm). Level 2b: identification by accurate mass (with an accepted deviation of 3ppm), and MS/MS spectra from in-house authentic standards and mzCloud (ThermoScientific) (match factor threshold >60), Level 3: identification by accurate mass alone (with an accepted deviation of 3ppm), based on the Human Metabolome Database (v. 5.0)[137]. Polar and semi-polar profiling results were merged in annotation levels 1 and 2a using compound annotations for matching. Duplicate detections based on similar retention time, accurate mass and signal correlation were removed manually. In the case of known isotopes with indistinguishable retention time and fragmentation behavior, co-eluting peaks were manually integrated and reported as compound groups (e.g. hexoses or sugar alcohols). Only compounds passing the following criteria were included in the results: (1) Precision (calculated as relative standard deviation between pooled QC samples) <20%, (2) Signal-to-noise (calculated as the ratio

between average(pooled QC) / average(blank) >5, (3) Correlation between signal and dilution factor in diluted QC's >0.8, and (4) Average(QC) / average(sample) between 0.5 and 2.

## Cell fractionation

At 70 % confluency one 10 cm dish of each of the NAT16 over-expressing cell lines was harvested by cell scraping. A centrifugation step at 17,000 x $g$ at 4 °C was performed to obtain cell pellets. To separate the different subcellular compartments and organelles, cell fractionation was performed based on a protocol from Baghirova et al.[138]. The pellet was resuspended in 500 µl lysis buffer A (pH 7.4, 150 mM NaCl, 50 mM HEPES, 25 µg/ml Digitonin, 1 x PIC) and incubated for 10 min at 4 °C in a rotating wheel. The whole lysate was separated into two fractions; one was kept for Western blot analysis, whereas 120 µl was centrifuged at 2000 x $g$ for 10 min at 4°. The supernatant was collected as the cytosolic fraction whereas the residual pellet was washed twice in 300 µl ice cold 1xPBS. To recover the pellet, centrifugation at 2000 x $g$ for 8 min at 4 °C was performed. Furthermore, the pellet was lysed in 120 µl lysis buffer B (pH 7.4, 150 mM NaCl, 50 mM HEPES, 1% NP-40, 1x PIC) on ice for 30 min before centrifugated at 7000 x $g$ for 10 min at 4 °C. The supernatant was collected as the organellar fraction. To isolate the nuclear fraction of proteins, the remaining pellet was washed as above before it was lysed in 120 µl of lysis buffer C (pH 8, 50 mM Tris-HCl, 150 mM NaCl, 5 mM EDTA and 0.5 % NP-40) on ice for 30 min.

All the fractionation samples were diluted in 6 x SDS-PAGE sample buffer and boiled for 5 min. The whole lysate and the nuclear fraction were additionally sonicated in intervals of 30 s x 4 for greater solubility. Lastly the fractions were analyzed by standard Western blotting setups. The samples were run on a 4–20 % Stain-Free Precast gel for 5 min on 100 V followed by 30 min on 200 V. The membrane was blocked with 5% dry milk for 1 h before incubated with antibodies overnight: Anti-V5 (Invitrogen, 46-1157, clone SV5-Pk1, lot 3045095, dilution 1:1000), Anti-Histone H4 (Millipore, 05-858, clone 62-141-13, lot not known, dilution 1:3000), Anti-Calnexin (Abcam, ab133615, clone EPR3633(2), lot 1036718-18, dilution 1:500) and Anti-GAPDH (Santa Cruz, Sc-47724, clone 0411, lot G2920, dilution 1:1000) which were diluted in 1 % dry milk. The secondary antibodies used were Anti-Rabbit (Cytiva, GENA934-1ML, lot 14187089, dilution 1:3000) and Anti-Mouse (Cytiva, GENA931-1ML, lot 17162269, dilution 1:3000) diluted in 3 % dry milk.

## Reporting summary

Further information on research design is available in the Nature Portfolio Reporting Summary linked to this article.

# Data availability

The crystal structures reported in this manuscript have been deposited to Protein Data Bank with identifiers 9EMD (NAT16 d5-27_2 with histidine and CoA disulfide), 9EMO (NAT16 d5-45_9 with arginine and CoA disulfide), 9EMP (NAT16 d5-45_18 with myristoylhistidine and CoA), 9EMT (NAT16 d5-27_42 with imidazole and CoA disulfide), and 9EN3 (NAT16 d5-45_49 with histidine and S-ethyl-CoA). These were compared to previously published structures 5O9U, 3C26, 2WPX, 2NMT, 3DDD, 7WX7, 1OZP, 7PK0, 3R1K, and 3SXN. The enzyme kinetics assay protocols and results have been deposited to STRENDA database with registry number HYQ0CA (WT) and ZT1UDP (F63S). The metabolomics data has been deposited to MetaboLights database with the identifier MTBLS12070. Source data are provided as a source data file.

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

## Acknowledgements

This work was supported by a grant from the European Research Council (ERC) under the European Union Horizon 2020 Research and Innovation Program (Grant 772039 to T.A.). We acknowledge the use of the Core Facility for Biophysics, Structural Biology, and Screening (BiSS) at the University of Bergen, which has received infrastructure funding from the Research Council of Norway (RCN) through NORCRYST (grant number 245828) and NOR–OPENSCREEN (grant number 245922). The cell sorting was performed at the Flow & Mass Cytometry Core Facility, Department of Clinical Science, University of Bergen. We acknowledge Deutsches Elektronen-Synchrotron (DESY, Hamburg, Germany), a member of the Helmholtz Association HGF, for the provision of experimental facilities. Parts of this research were carried out at PETRA III, and we would like to thank Dr. Johanna Hakanpää and Dr. Eva Crosas for assistance in using the P11 beamline. Beamtime was allocated for proposals BAG-20190768 EC and BAG-20211049 EC. Some of the synchrotron data was collected at the P14 beamline operated by EMBL Hamburg at the PETRA III storage ring (DESY, Hamburg, Germany). We would like to thank Dr. Gleb Bourenkov for the assistance in using this beamline. This research was supported in part through the Maxwell computational resources operated at DESY, Hamburg, Germany. We would like to thank Dr. Petri Kursula for the data collection of the iodide derivative datasets.

## Author contributions

M.M., M.L., C.O. and S.S.N. conducted experiments. M.M., M.L., C.O., and T.A. analyzed data. M.M., M.L., C.O. and S.S.N. prepared figures. M.M. and T.A. wrote the manuscript with inputs and revisions from all authors.

## Funding

## Competing interests

The authors declare no competing interests.
