## [Transparent Peer Review file · Nature Communications]

The molecular basis for acetylhistidine synthesis by HisAT/NAT16

Corresponding Author: Dr Matti Myllykoski

Version 0:

Reviewer comments:

Reviewer #1

(Remarks to the Author)

[Editorial Note: The reviewer's attachment is displayed at the end of this file]

Reviewer #2

(Remarks to the Author)

The reviewer was only tasked to review technical details of the metabolomics analyses.

The following points arise:

(a) data are not uploaded to public repositories, e.g. MetaboLights or MetabolomicsWorkbench. Authors must upload both raw instrument files and result data. The authors make a false statement when they declare in the NR reporting summary that all other data are included with the manuscript. The metabolomic data has not been included in either the manuscript or the supplement. The supplement only includes compound names, RT and m/z but not MS/MS information, and not the library that was used for compound ID and MS/MS similarity thresholds. The supplement does not include the actual data, but only the statistics results. The supplement does not include the instrument raw data, either.

(b) cells were extracted by methanol/chloroform (80:20), excluding highly hydrophilic compounds such as sugar acids. This is probably fine with the aim of the analyses that looked at amino acids, but should be discussed as a limitation. Given the very lipophilic extraction method, it is strange to see that the authors did not detect lipids in their 'semi-polar' chromatography method.

(c) The authors do not disclose the chromatography methods, but refer to two former publications, and even say that methods were modified ("using a slightly modified version of the protocol described by Hsiao et al. 132 for polar metabolites or by Doneanu et al. for semi-polar metabolites."). This is not acceptable reporting practice. Methods must be given in detail to allow other researchers to repeat analyses. Method details can also be given in supplements, if needed, and if method sections get too long otherwise.

(d) Compound annotations are likely correct, and authors give some criteria (but not RT windows and not MS/MS similarity thresholds). Software Compound Discoverer is fine, but the authors do not report the confidence levels reported by that software, and authors do not report the parameters used for data processing. A number of compounds are not fully annotated ('hexose' or 'sugar alcohol'), and it is unclear which external library was used for compound annotations.

Reviewer #3

(Remarks to the Author)

The paper by Myllykoski et al provides new, important information on Nat16, as the enzyme that acetylates histidine in specific cells and tissues in humans. Until now, this function had only been described in a fish, and the human recombinant enzyme had been tested, but was apparently without any activity. The function is now clear and the role of Nat16 needs to

be found. Nat16 is present in many vertebrates and in more primitive organisms, but it is apparently absent from most rodents, though, as stressed by the authors, the difficulty of sequencing Nat16 in many species, makes that it is difficult to be sure about this absence. The paper provides a quite complete kinetic investigation of Nat16 and provides its structure, with its peculiar double GNAT fold, which is compared with that of other enzymes of the GNAT family.

Overall the paper provides a lot information and provides a good coverage of the literature. This paper is a major step in the elucidation of the function of N-acetylhistidine, which is predominantly expressed in brain and secretory tissues. The experiments have been carefully performed and well analysed.

I have only minor comments

Summary Line 22 ...Lower blood levels of acetylhistidine which impacts kidney function

This a strong statement: nothing is known on how a reduced acetylhistidine level could cause kidney dysfunction.

Figure 1. Arginine and lysine are substrates for Nat16, as shown in Fig1A. Since the results in Fig. 1a underestimate the activity on histidine, it would be good to provide values of K_m and V_{max} of the enzyme when lysine or arginine are used as a substrate. There is no need to perform as detailed a kinetic investigation as performed for histidine.

Line 117. Please provide information on the concentration of histidine used for determining the kinetic constant in Fig. 1C

Figure 2C : overexpression of Nat16 seems to cause higher increases in acetyllysine than in N-acetylarginine, yet arginine seems to be a better substrate on the purified enzyme than lysine. What does 'relative abundance' mean ? Are the values for the different metabolites comparable ?

Line 609, page 25; were the kinetic studies performed with a whole set of Acetyl-CoA concentrations combined with a whole set of histidine concentration, as the sentence suggests ?

Supplementary figure 2, legend; please indicate the concentration of histidine that was used

Version 1:

Reviewer comments:

Reviewer #1

(Remarks to the Author)

Response to the authors' comments to the manuscript entitled "The molecular basis for acetyl-histidine synthesis by HisAT/NAT16" submitted by Myllykoski and co-authors.

The manuscript did strongly improve in this round of revision. I do not have any further open points which need to be addressed by the authors as you see on my comments (in red) to the answers supplied by the authors. Overall, it is a highly interesting study and I recommend publication of this manuscript in its current form.

[Editorial Note: The reviewer's attachment is displayed at the end of this file]

Reviewer #2

(Remarks to the Author)

the authors have uploaded raw files and method descriptions to the MetaboLights repository. The reviewer has not assessed the veracity of claims for compound IDs, or the usability and information content of the files, but broadly, uploads appear to be accessible.

Reviewer #3

(Remarks to the Author)

I Think that this paper is an important report on the enzyme that acetylates histidine. Though the function of N-acetylhistidine remains mysterious and though the enzyme that makes it seems to be missing in many species, the paper is a good start to understand what this function may be and if there human diseases may result from its deficiency (or its overproduction).

The authors answered to my questions in a satisfactory manner.

Rebuttal letter for “*The molecular basis for acetyl-histidine synthesis by HisAT/NAT16*”.

We thank the Editor and reviewers for their useful inputs to our work and for the invitation to resubmit an improved manuscript version. We have carried out additional experiments and adjusted the manuscript text and figures according to reviewer comments, and we hope that the revised manuscript is found acceptable for publication. Please see specific actions and responses to reviewer comments below.

Reviewer #1 (Remarks to the Author):

Review to the manuscript entitled “*The molecular basis for acetyl-histidine synthesis 1 by HisAT/NAT16*” submitted by Myllykoski and co-authors

The manuscript entitled “*The molecular basis for acetyl-histidine synthesis by HisAT/NAT16*” submitted by Myllykoski and co-authors describes the structural and functional characterization of the GNAT-acyltransferase NAT16 as N-(a)-histidine acyltransferase. The authors performed enzyme kinetics applying an DTNB assay to detect the thiol groups of released coenzyme A to study if NAT16 uses proteinogenic amino acids and further substrate candidates as substrates. These pre-screening studies indicated that NAT16 is capable to use histidine as major substrate but that it is also capable to use arginine, lysine and the aromatic amino acids phenylalanine and tyrosine as substrates. The authors confirmed these *in vitro* studies by performing metabolomics studies from haploid HAP1 cells generating a *Dnat16* knockout cell line to show that the knockout has an impact on the acetylhistidine level while not affecting the acetylation level of other amino acids. The authors show that the enzyme NAT16 is localized in the cytosol and conclude from a public database that the expression is high in retina, kidney and brain. Moreover, the authors conduct Michaelis-Menten kinetics and propose these data suggest an enzymatic mechanism involving formation of a ternary complex between enzyme, substrate, i.e. histidine, and co-substrate, i.e. acetyl-CoA, and in which an tetrahedral intermediate is formed. The authors characterize a mutation of NAT16, i.e. F63S, which was shown to result in an decrease in blood serum acetyl-histidine, which correlates with certain forms of kidney disease. Biochemical characterization show this mutant shows a slightly increased K_M value suggesting that it interferes with binding affinity towards histidine while not affecting catalysis. Finally, the authors present several crystal structures of NAT16 in complexes with histidine and CoA-SS-CoA disulfide, arginine and CoA-SS-CoA disulfide, imidazole and CoA-SS-CoA disulfide, histidine and ethyl-CoA, and myristoyl-histidine and CoA. The authors provide a characterization of the substrate binding site, suggest potential residues involved in formation of the oxyanion hole during catalysis but do not present data on a potential general base. The authors show that NAT16 is structurally related to N-terminal myristoyl-transferase 1/2 (NMT1/2) as both have a double GNAT-domain architecture. The phylogenetic data/dendrogram should show some evolutionary development of NAT16 and NMT1/2 but is a bit confusing. Overall, the physiological significance of the enzyme NAT16 is not resolved and the role of acetylation of histidine side chains is unclear. In total the manuscript is important and gives novel insights into acetyltransferases with direct correlation to human diseases. To this end, I recommend in principle the publication of the manuscript but certain issues need to be resolved prior to publication as explained in the next section.

Open Points:

1. There is a class of bacterial GNAT enzymes, i.e. type V GNATs, which also encompass a double-GNAT domain which is followed by a C-terminal domain. Does NAT16 show similarities, i.e. in terms of biochemistry, structure and function, to these type V GNATs?

RESPONSE: Type V GNATs are a group of bacterial enzymes of which the *Mycobacterium tuberculosis* enzyme Eis (Enhanced intracellular survival) seems to be the best characterized. Eis is a secreted enzyme that acetylates a broad range of substrates including aminoglycosides, lysine side chains of peptides, and arylalkylamines such as histamine. Eis type enzymes consist of two GNAT domains arranged in a similar way to NAT16/HisAT and an additional C-terminal domain with homology towards sterol carrier proteins (Chen et al. 2011, PMID: 21628583).

Structurally Eis and NAT16/HisAT are similar in that they both have the double-GNAT fold and for both the N-terminal of the two GNAT domains is the active one. They are different in that Eis has the additional C-terminal domain, and that NAT16/HisAT appears to be a monomer while Eis has a conserved hexameric tertiary structure. Eis has also a more open substrate binding cavity between the GNAT domains, possibly reflecting the broader substrate range. The carboxyl group of the very C-terminal residue of Eis enters the active site and is involved in catalysis in the same way as with NMTs but unlike with NAT16/HisAT, and Eis does not have a corresponding β -hairpin that we used to identify NAT16/HisAT type structures, suggesting Eis structure is closer to the NMT-type than the HisAT-type.

Eis has broad activity towards different aminoglycosides often acetylating multiple amides of one molecule, lysine side chains of peptides, and even arylalkylamines such as histamine (Chen et al. 2011, PMID: 21628583, Kim et al. 2012, PMID: 22547814, Pan et al. 2018, PMID: 29402941), while NAT16/HisAT appears to be more focused with activity towards histidine and residual activity towards other amino acids. Eis enzymes may also be more limited in their ability to accept longer acyl-CoA substrates (Green et al. 2015, PMID: 27622743). As far as we can tell, it is not known if Eis enzymes can acetylate histidine, but given their broad substrate specificity, and the capacity to acetylate histamine, it would not be very surprising.

Functionally, the secreted *M. tuberculosis* Eis is thought to confer resistance against aminoglycoside antibiotics such as kanamycin and other anti-TB drugs by acetylating them, and to enhance the survival of the bacteria within macrophages by the activation of DUSP16 via its acetylation at lysine 55. NAT16/HisAT is an intracellular cytosolic histidine acetyltransferase, but the roles of this activity and its product are currently unknown.

We initially overlooked this group of enzymes, as they didn't score highly in similarity searches possibly because of the extra domain. We have now mentioned them in the introduction (line 74) and discussion (line 518). In addition, we repeated the Dali structural similarity analysis with two mycobacterial Eis proteins included in a revised Figure 6a. The placement of the Eis proteins in the resulting dendrogram as early branches of the NMT-/HLS-type half of the dendrogram appears to confirm our suggestion above that they are structurally closer to NMT-type than HisAT-type.

2. Avoid emotional or judging language such as “overwhelmingly” in line 91, “good” in line 92, “worse” in line 257. Describe it scientifically. What do you precisely want to say with “good”, what do you define as “worse”, etc.

RESPONSE: We have removed or rephrased these words.

3. Line 95: write min instead of minutes and add the unit also to the number 30.

RESPONSE: We modified the text as suggested. (Line 96)

4. It is not clear from the manuscript if NAT16 has any peptide or protein substrates. The authors describe that they tested various peptides which were shown in a table in the SI but these peptides do not contain an N terminal His. The authors should test if NAT16 is capable to acetylate peptides or proteins with an N-terminal His. This would have further implications for the function of the enzyme modulating peptide/protein function and processes such as protein stability/turnover. Along that line could the increase in blood acetyl-lysine also be due to degraded proteins that carried an acetyl-histidine at the N-terminus rather than due to acetylation of isolated histidine side chains?

RESPONSE: As far as we know, NAT16 does not have peptide or protein substrates. We have now tested selected His-starting peptides (together with Ser-starting controls) that have been reported to exist in cells and some of which were reported to be acetylated (Yeom et al. 2017, PMID: 28747677). There was no detectable activity towards these peptides, see revised Supplementary Table 1 and comment in the Results, (line 100). Considering the clear acetyltransferase activity towards histidine and no activity towards peptides with histidine or other termini, we do not believe the proposed mechanism is a likely source of acetylhistidine.

5. The authors performed metabolomics to show that knockout of NAT16 resulted in a decrease in acetyl-lysine levels. Furthermore, they state that for lysine it is not clear if lysine is acetylated at the side chain, i.e. N-(ε)-acetylation, or at the N-terminus, i.e. N-(α)-acetylation. Can the authors explain why they can be sure that histidine is not acetylated at the δ- or ε-N of the imidazole ring? If it is in its unprotonated state it might be competent to attack as a nucleophile the electrophilic carbonyl carbon of the acetyl-group of acetyl-CoA. Histidine side chains can also be phosphorylated and these side chains nitrogens. Can the authors show this without any doubt, i.e. does the MS/MS measurement result in fragmentation spectra.

RESPONSE: There was no difference in acetyllysine levels in NAT16 knockout vs. WT cells (without NAT16 overexpression).

The metabolomic analysis identified acetylhistidine at level 1 based on three criteria: identification by retention times (compared against in-house authentic standards +/-0.2min), accurate mass (with an accepted deviation of 3ppm), and MS/MS spectra (match factor threshold >60). The first of these was missing for the acetyllysine metabolite, making its reliable identification less certain. There are also multiple additional lines of evidence pointing towards N-alpha acetylation. A methylated nitrogen would be expected to block the acetylation of the same nitrogen in the histidine side chain, but both methylhistidines were acetylated similarly to the unmethylated histidine. In the crystal structure the alpha-amino group of histidine is positioned right next to where the acetyl group of Ac-CoA would be, while the side chain nitrogens are further away. Finally, there are numerous reports in the literature reporting N-alpha-acetylhistidine being detected in animal tissues. The reports of side chain acetylated histidines are very rare.

6. The authors show binding of a oxidized double-CoA molecule, i.e. two CoA molecules linked together covalently by forming a disulfide bond via their thiol groups. Do the authors think that both binding sites are important for the activity of the enzyme? Are both saturated in their enzyme kinetics? How are the affinities for both sites? It might play an important regulatory role if for example the second CoA-binding site has a lower affinity compared to the catalytic acetyl-CoA binding

suggesting that it might impair enzyme activity at very high CoA concentrations. Along that line, would it be competent structurally to accommodate two acetyl-CoA molecules or is binding of the CoA-SS-CoA only possible with two CoA molecules or is it competent to bind one acetyl-CoA and one CoA? Along that line, the authors say that they observe significant substrate inhibition at higher acetyl-CoA concentrations. I do not see the substrate inhibition in the data (Supp. Fig. 2). Where is it shown? Can the mechanism underlying the mode of substrate inhibition have sth. To do with saturating bot CoA binding sites? Maybe there is some CoA in the acetyl-CoA stock, it is quite unstable.

RESPONSE: We think that only the canonical Ac-CoA binding site is required for catalysis. We have not measured the affinities of the different binding sites. It is not clear to what extent the second site is (Ac-)CoA binding site and to what extent it is an artefact of crystallization. We only managed to produce the P63 crystals when both CoA molecules were present, and the second one was generating a crystal contact. Further, it may be that the C-terminal 6xHis tag that was part of this crystal contact, is also required for its binding. There was no electron density in the second CoA binding site in the structure with S-ethyl-CoA even though the concentration for this substrate analog was 0.7 mM in the crystallization buffer and 1 mM in the cryoprotectant soaking solution, suggesting that two Ac-CoA molecules probably cannot bind simultaneously.

We identified the substrate inhibition from the decrease in the limiting rate (V_{max}) in the Michaelis-Menten curves in Supplementary Figure 2a and the values listed in Supplementary Figure 2d, with higher Ac-CoA concentrations. For example, in series 2 and 3, the reactions with 500 μ M Ac-CoA have similar limiting rates as the reactions with 12 μ M Ac-CoA, while the highest rates are seen with reactions containing 30 μ M Ac-CoA. Please see also the response to comment 10 and the new Supplementary Figure 2e.

The mechanism of substrate inhibition could potentially be the result of Ac-CoA or CoA binding to the second site preventing productive binding to the canonical site, but there is no solid evidence for the inhibitory mechanism at present. Cellular Ac-CoA levels are typically reported to range from low micromolar to few tens of micromolar. At those Ac-CoA levels the inhibitory properties may be irrelevant. We assumed that any CoA derived from the Ac-CoA stock present in the reaction mixture before enzyme addition would react with the DTNB present and not be available for product inhibition.

7. Fig. 1 A: please show mean values and standard deviations (and not just their “average”) and write in the figure legends how often you repeated the experiments and which statistics tests were performed. How did you determine the CoA-concentrations? Did you generate a calibration curve? Can you show it in the SI section.

RESPONSE: We have modified Figure 1a to show the individual data points, their mean, and the standard deviation as instructed in this comment and the editorial guideline. We have modified the legend as instructed. The CoA concentrations were determined using Beer's law using the pathlength correction described in the reference 107 and the extinction coefficient $13700 \text{ M}^{-1} \text{ cm}^{-1}$. In this pathlength correction procedure, absorbance at 900 nm is subtracted from absorbance at 975 nm and the result is divided by a constant (0.186) to obtain the pathlength in cm.

8. Fig. 1B: can you not draw chemical structures yourself instead of adapting it from a different source? Why do you show different His tautomers following acetylation? Is the tautomer protonated at N-d favored if histidine is acetylated?

RESPONSE: We have drawn the structures as instructed in a revised Figure 1b. We do not think the side chain tautomer is relevant for the acetylation.

9. Fig. 1C: Can the authors explain where these numbers come from? Maybe show here the mean values and standard deviations. Also show the k_{cat}/K_M values as this is the value that should be used to compare the enzymes' efficiencies.

RESPONSE: We have modified Figure 1c as suggested. The K_M for Ac-CoA was derived from the negative inverse of the X-intercept in Supplementary Figure 2d. The same plot was used to derive V_{max} (not shown here) from the inverse of the Y-intercept. The Y-intercept of the plot in Supplementary Figure 2e corresponds to the K_M for histidine divided by V_{max} , so the K_M for histidine was obtained by multiplying this Y-intercept with V_{max} . The k_{cat} values were calculated from the V_{max} values by dividing with the enzyme concentration (0.02 μM). The k_{cat} / K_M values were calculated from the k_{cat} and K_M values.

10. Supp. Fig. 2: The authors show the Michaelis-Menten kinetics. In this case the enzyme uses two (co)substrates, i.e. histidine and acetyl-CoA. In panel A the authors show three series of MM-kinetics. Can the authors not also show the values for the higher acetyl-CoA concentrations to see the effect of substrate inhibition? The authors could fit the data with a model "Michaelis-Menten kinetics with substrate inhibition" which would also yield the K_i for the substrate inhibition:

$$V_0/[E]_0 = k_{cat}[S]/(K_M+[S]/(1+[S]/K_i))$$

RESPONSE: The data points with Ac-CoA up to 500 μM , which is the highest concentration we used, are already included in Figure 2a, and there is a clear decrease in the limiting rate at higher Ac-CoA concentrations. We have included the $v_0 = V_{max} [\text{Ac-CoA}] / (K_M + [\text{Ac-CoA}] (1+[\text{Ac-CoA}] / K_i))$ substrate inhibition plot for the Series 3 in Supplementary Figure 2e. The mean of the K_i from the curves at different histidine concentrations was 333 μM (St.Dev. 19 μM).

11. The authors derive from the data shown in Supp. Fig. 2B that the enzyme uses a sequential mechanism, i.e. formation of a ternary complex, and furthermore that formation of a tetrahedral intermediate is obvious. If you have an enzyme with two substrates it is correct to perform MM-kinetics varying both concentrations as done here. Panel A suggests that all MM-kinetics will result in the same (or highly similar K_M -value for [His]). However, in Fig. B there is no Lineweaver Burk plot shown, which would show $1/V_0$ at the y-axis and $1/[S]$ on the x-axis, but this is not the case. If this would be the case and the lines would cross the x-axis (and not the y-axis as written in the figure legend!) in the same point, it would indicate a sequential kinetics. I do not get the point why the authors concluded from these data that the mechanism proceeds via a tetrahedral intermediate.

RESPONSE: As is correctly stated here, the plot in panel B is not a double-reciprocal (Lineweaver-Burk) plot, but an a/v vs. a plot, sometimes called Hanes plot or Hanes-Woolf plot. Its capability to differentiate between ternary complex and substituted enzyme mechanisms is derived from the same source as it is for the double-reciprocal plot, namely the presence of the inhibitory term K_{iA} in the denominator of the rate equation for the ternary complex mechanism but not for the substituted enzyme mechanism. This leads to the easily discernible behavior of the two mechanisms when plotted on the a/v vs. a plot: In a plot for ternary complex mechanism the lines representing data points measured at different concentrations of substrate B will intercept each other at the X-axis value of $-K_{iA}$, while in a plot for the substituted enzyme mechanism they will intercept at the Y-axis ($X=0$), because the term K_{iA} is not present in the rate equation. Thus, as the lines of the plot on panel B

intercept to the left of the Y-axis (at a negative value) and not on the Y-axis, we concluded that the enzyme utilizes a ternary complex mechanism.

On a double-reciprocal plot, a substituted enzyme mechanism produces lines that are parallel, while the lines representing data points measured at different concentrations of substrate B produced by a ternary complex mechanism are not parallel. A double-reciprocal plot using the same underlying data as on Supplementary Figure 2b (and the first panel of Supplementary Figure 2a) is now included as Supplementary Figure 2c. This plot also indicates a ternary complex mechanism.

12. For GNATs reports are available that confirm and support the sequential kinetics mechanism. These should be cited.

RESPONSE: We have discussed the GNAT family enzyme mechanisms in the discussion section (lines 514-516).

13. MM-kinetics: it would have been straightforward to keep one concentration constant (at saturating level) and just change the other concentration to determine K_M and k_{cat} for one (co)substrate and do it vice versa for the other substrate.

RESPONSE: Using this approach there would be no information about the kinetic mechanism. Furthermore, using saturating concentration becomes complicated in the presence of substrate inhibition.

14. Supp Fig. 1C/D: Please explain exactly what is plotted here and where the data were obtained from that were used for these plots.

RESPONSE: Supplementary Figure 1c is a picture of the Coomassie-stained SDS-PAGE ran with the fractions from the size exclusion chromatography depicted in Supplementary Figure 1b.

Supplementary Figure 1d plots the time course of NAT16 DTNB assay to determine how rapidly NAT16 consumes all Ac-CoA in the reaction mixture of the screening assay. The plot consists of concentrations of CoA, from which a control tube signal was subtracted, as a function of time. CoA concentration was determined as described in the methods and the underlying data are included in the source data file.

15. Fig. 1C: The K_M for His can be obtained from the MM-kinetics plots shown in Supp. Fig. 2A. Can the authors explain how they obtained the K_M values for acetyl-CoA and how they determined the k_{cat} values shown in 1C.

RESPONSE: The K_M and V_{max} values, derived from the fitting of the Michaelis-Menten plot in Supplementary Figure 2a at fixed Ac-CoA concentrations, were considered the apparent K_M and V_{max} values. As described in Cornish-Bowden (2013, Fundamentals of Enzyme Kinetics, Wiley-Blackwell), the apparent V_{max} (V_{max}^{APP}) determined by varying substrate A (histidine) at different fixed substrate B (Ac-CoA) concentrations, can be described by the equation $V_{max}^{APP} = (V_{max} [Ac-CoA]) / (K_M^{Ac-CoA} + [Ac-CoA])$. As this equation is of the form of the Michaelis-Menten equation, V_{max} and K_M^{Ac-CoA} can be determined from the double-reciprocal secondary plot of $1 / V_{max}^{APP}$ vs. $1 / [Ac-CoA]$ generated using the apparent V_{max} values derived from the fits of Supplementary Figure 2a and listed in Supplementary Figure 2d. This plot is shown in supplementary figure 2f (it was panel 2c in the initial submission). V_{max} was derived from the inverse of the Y-axis intercept and K_M^{Ac-CoA} from the negative inverse of the X-axis intercept. The k_{cat} values were calculated from the V_{max} values by dividing with the enzyme concentration.

We have clarified the procedure in the methods (line 692-708) and in the Supplementary figure 2 legend.

16. Supp. 1D: which data were derived from Supp. Fig. 2D. Please explain to be able to follow it.

RESPONSE: Supplementary Figure 2d in the initial submission is the panel 2g in current version of the manuscript.

Supplementary Figure 2g was used to determine K_M for histidine. As described in Cornish-Bowden (2013, Fundamentals of Enzyme Kinetics, Wiley-Blackwell), the ratio of the apparent constants $V_{max}^{App} / K_M^{App}$ determined by varying substrate A (histidine) at different fixed substrate B (Ac-CoA) concentrations, can be represented in Michaelis-Menten form by the equation $V_{max}^{App} / K_M^{App} = ((V_{max} / K_M^{His}) [Ac-CoA]) / (((K_i^{His} K_M^{Ac-CoA}) / K_M^{His}) + [Ac-CoA])$. Thus, another double-reciprocal secondary plot (Supplementary Figure 2g) of $K_M^{App} / V_{max}^{App}$ vs. $1 / [Ac-CoA]$ can be used to determine, from the Y-intercept, K_M^{His} / V_{max} , from which K_M^{His} was calculated by multiplication with V_{max} derived from Supplementary Figure 2f as described above.

We have clarified the procedure in the methods (line 692-708) and in the Supplementary figure 2 legend.

17. MM-kinetics: show also the primary data, i.e. [CoA] as a function of time.

RESPONSE: These data are included in the source data file.

18. Line 138, "...assay did not reveal the location of the acetyl group": please explain better what you want to say here, i.e. either acetylation at N-(a)- or N-(e) of lysine side chain.

RESPONSE: We have modified the text here as suggested (line 152-153).

19. Line 132, "...was significantly decreased": did the authors perform a statistics test? If yes show it and say in the legend what test was conducted and show the significance level.

RESPONSE: The details about the statistical tests are now included in the legend of the Figure 2 and in supplementary table 2. The statistical significance levels are indicated in Figure 2.

20. Fig. 2: Write "Western" with capital "W" as the name is derived from a surname.

RESPONSE: We have modified the text as suggested.

21. Fig. 2C: Can the authors judge how the stoichiometry of acetylation on histidine is. As shown by the metabolomics, they identified histidine and acetyl-histidine in the cells. Were both present at similar amount/concentration? Can the authors conclude anything concerning the stoichiometry?

RESPONSE: The relative abundances of the different metabolites are not comparable in this assay, so we cannot determine the relative stoichiometry of histidine vs. acetylhistidine. We have clarified this in the figure legend.

22. Line 165, "This agrees with a model in which": can the authors explain which model they refer to and cite the reference.

RESPONSE: The model we refer to was described in the same sentence after the word 'which', briefly, that the acetylhistidine produced by HisAT-containing cells is secreted to blood by the endocrine tissues. To clarify this, we have changed the wording of this sentence (line 188-189).

23. Lines 174/175: Please show a clear image of the structure from which the location of the mutation F63S becomes clear. Is it near the CoA binding site? Is it near the His binding site?

RESPONSE: We have removed the structure from Figure 3c and included a more detailed image of Phe63 in Supplementary Figure 5g. We refer to the figure and the position of the residue later when we discuss the structure (Line 259).

The residue is not very close to the active site (shortest distances to the substrate and co-substrate 14 and 15 Å, respectively), but it is located in the $\alpha 1$ helix and as the $\alpha 1$ - $\alpha 2$ loop is involved in substrate binding, it could be that mutation to serine somehow alters the orientation of this loop.

24. Lines 177-180: the authors show that the mutation F63S in NAT16 affects the K_M value and only slightly k_{cat} . They state that an effect is observable "especially at low histidine conditions". It is important to note, that the concentrations matter, so replace "low histidine conditions" with "low concentrations of histidine". If the enzyme is saturated with histidine no effect of the mutant is detectable as k_{cat} is almost unchanged. In this respect, is it known how the intracellular concentrations of histidine are in the tissues where NAT16 is expressed. Is a difference of the activity of the mutant compared to wildtype expectable? This will help to judge the physiological importance of the mutant. This could be discussed.

RESPONSE: We have modified the text as suggested.

We have found it difficult to find reliable numbers for free histidine concentration in human tissues. The range 70-120 μM has been suggested for blood histidine concentration (Brosnan & Brosnan 2020, PMID: 33000155). Hu et al. (2017, PMID: 28252043) used a fluorescent probe to determine histidine level in glucose-fed and -starved HeLa cells in culture and found the cytosolic histidine to be 159 μM in fed and 20 μM in starved cells. Using the apparent K_M values for WT vs. F63S HisAT, the starved cell histidine concentration would result in roughly 3-fold slower rate for the mutant enzyme, while using the fed cell histidine concentration, the difference is 2-fold. However, we do not know how well these histidine levels match the levels in intact tissues or different types of cells, and this kind of kinetic calculation is a gross simplification of the cellular situation. We have mentioned these rate differences in the discussion (lines 505-506).

25. Fig. 3A,B: this is a figure derived from a public database. I suggest moving it into the SI section.

RESPONSE: As suggested, these figure panels have been moved to Supplementary Figure 4.

26. If seeing Fig. 3A/B the question is if the differences in the mRNA levels are manifested on the protein level in the tissues.

RESPONSE: Unfortunately, such data on tissue expression of the NAT16 protein is not available.

Furthermore, we are somewhat skeptical about the specificity of several of the commercially available NAT16 antibodies as they detect multiple bands on immunoblots but only detect NAT16 when it is overexpressed.

27. Fig. 3C: the structural models are not really helpful to see where the mutation is located. Is it an AlphaFold model or which structure is used here? The only difference in the two subpanels is the color. I do not see a clear benefit showing this.

RESPONSE: We agree and we have removed the structures from this panel.

28. Fig. 3C: the bar graphs lack correct labels on the y-axis, i.e. label and unit.

RESPONSE: We have modified the graphs to include correct labels on the Y-axis.

29. Check throughout if all graphs have correct labels on the axes; there are graphs in the main section and in the SI section that lack properly labelled axes.

RESPONSE: Graphs have been checked and adjusted.

30. Just for curiosity, can the author explain why they used isomorphous replacement to solve the phase problem and not molecular replacement? AlphaFold2 generates excellent molecular replacement models or did it not work out for the authors?

RESPONSE: The first of the NAT16 structures discussed here was solved in October 2020. At the time Alphafold2 was not publicly available. Structure predictions at the time were not of sufficient quality to help solve the crystal structure with molecular replacement. The predictions were especially poor for the second GNAT domain that we did not identify as having a GNAT-like fold before we solved the structure.

31. Line 202: “coenzyme A disulfide molecule”: the authors should explain a bit better, what they want to say. i.e. it is two CoA covalently connected by formation of a disulfide bridge with their thiols.

RESPONSE: We have clarified the text (line 234-235).

32. Line 203: “crystallized also in space group I222”: why do the authors say “also” here; it is the first time they mention this space group in the manuscript. That is a bit confusing.

RESPONSE: This is because the shorter construct was crystallized in both space groups. We have modified the text to clarify this (line 236-237).

33. Line 204: “the truncated N-terminus interacted”, Line 206: “whole truncated N-terminus was visible”: these statements are really confusion. How can a part that is truncated, i.e. not present, interact with sth. or can be visible. Please work on the language and explain it differently.

RESPONSE: We have added amino acid information to improve clarity. Truncated does not mean “not present”, but “cut short” or “lacking an expected or normal element at the beginning or end” (<https://www.merriam-webster.com/dictionary/truncated>). The language refers to the N-terminus after the removal of residues 5-45 and it is correctly used here.

34. Show a 2Fo-Fc map for all ligands.

RESPONSE: We have generated a supplementary figure (supplementary figure 7) with 2Fo-Fc map for ligands.

35. Show a proper Fo-Fc omit map for all ligands and replace the polder map with those. A polder map is not a real omit map.

RESPONSE: We have replaced the polder maps with omit maps in supplementary figure 8.

36. All structure images: the images should be labelled better. Label all important parts, side chains, CoA, α -helices, β -strands, shown. Some figures are really crowded. Please work on the images to make it better readable.

RESPONSE: We have improved the clarity of all the structure figures as requested.

37. The ethyl-CoA is firstly mentioned in Fig. 4 and line 241 but it is explained later in lines 266 and following. Please shift this section to explain the compound when it is introduced.

RESPONSE: We have rearranged and modified the text as suggested.

38. Give information on how the structures with ligands were obtained, was it by soaking or by co-crystallization?

RESPONSE: The information about crystal growing and soaking solutions is included in Supplementary table 4.

39. Line 261: The authors state that differences in the affinities might explain the difference observed in enzyme activity between arginine and histidine and poor activity towards other amino acids. If so, the authors should observe activity under saturating concentrations of the amino acids.

RESPONSE: We have now determined the apparent kinetic parameters for HisAT also in the presence of arginine (Supplementary Figure 2h-i).

40. Line 257, "lower resolution cutoff": the cutoff is set by the user, applying certain rules where to set the cutoff for the resolution, i.e. $CC1/2$ (or I/sI , R_{meas}/R_{merge}).

RESPONSE: We have replaced this text with a reference to the diffraction data statistics table.

41. Line 258, "The cause for the HisAT substrate preference...". The authors explained that the imidazole ring can form quite some interactions with side chains and main chains in the enzyme's active site. Why does not explain the specificity. Aromatic side chains, Phe and Tyr, might form the stacking interactions with the nearby aromatic side chains in the active site, and Trp might just be too bulky to fit into the active site, i.e. I would sterically not be possible to be a substrate.

RESPONSE: The only direct interactions the imidazole side chain of histidine makes with the amino acid residues in the active site arise from being stacked in between Tyr74 and Trp246. The substrate histidine interacts with the active site with the α -amino and carboxyl groups, but these interactions would presumably be shared by all amino acids (except perhaps proline), and the stacking interaction could presumably be shared at least by tyrosine and phenylalanine. However, tyrosine and phenylalanine are not efficiently acetylated by HisAT. We propose here that the acidic nature of several residues bordering the active site could contribute to the preferred substrate being histidine, but this is only our best guess at this time, which is what we attempted to convey here.

42. Line 271, "These two amides generate an oxyanion hole...". Is that speculation or is that derived from structural or sequence alignments? Are these residues conserved in other GNATs? Do the authors show this in a figure?

RESPONSE: The presence of the β -bulge in most GNAT-fold enzymes is well known. The β -bulge residues forming an oxyanion hole is also well known (Vetting et al. 2005, PMID: 15581578, Salah-Uddin et al. 2016, PMID: 27367672), for example for NMT1 (Dian et al. 2020, PMID: 32111831). The residues themselves are not broadly conserved, but their orientation is. We have introduced these references here for clarity.

43. For the section of comparison of NAT16 with NMT1/" show a structural superposition and calculate R.M.S.D. values to get a value for their similarity.

RESPONSE: The superposition is shown in Fig. 5c and the RMSD is found in the figure legend. We have presented it in this way for all structure superpositions.

44. Line 304: Arg161 and Asp163 are not visible in the electron density of the substrate structure. Are they in flexible parts which are fixed upon myristoyl-histidine binding?

RESPONSE: The poor electron density concerns only the side chains, and they do still have some density. The side chain electron density appears worse when neither myristoylhistidine nor the second CoA molecule are present, suggesting they adopt more static positions in the crystal when a ligand is bound nearby. However, the difference isn't large and is only mentioned as for the most part the active sites compared here are very similar.

45. Line 318: "...that clearly contributes to enzymatic activity or substrate binding is Trp246". How do the authors conclude this? Is that based on the structures showing that it is structurally involved binding the histidine? Do the authors have MM-kinetics for the mutant?

RESPONSE: This statement is based on the structural data of the substrate binding by Trp246, and its conservation among different HisAT-type proteins. We do not have enzyme kinetics data of a mutant. We have modified the statement accordingly (line 360-361).

46. Line 326, "... and myristoylated peptide substrate in NMT1": if the binding of substrate so similar in both enzymes, maybe this is a hint for NAT16 also being able to act as peptide/protein N-(a)-acetyltransferase if a histidine is at the N-terminus an not only active on the isolated amino acid similar to other N-terminal acetyltransferases acting co- or posttranslationally. Maybe the efficiency is higher for a peptide/protein as other parts apart from the histidine side chain contribute to binding affinity.

RESPONSE: We have corrected an error in the above quotation, the myristoylated peptide is of course a product of NMT1.

The binding of the myristoylated products is only similar for the myristoyl chain, not for the peptide or amino acid part. As seen in Figure 5 c-e; histidine binding is incompatible with it being N-terminal in a polypeptide as the polypeptide chain in the NMT1 product extends to the direction of the histidine side chain. We have now also tested NAT16 activity towards several His-starting peptides and found there to be none (Supplementary table 1).

47. Lines 349 and following: Can the authors explain why the neighboring genes to *nat16* 169 are so important? Do the authors have indications that these are somehow connected on the transcriptional or post-transcriptional level?

RESPONSE: The neighboring genes were used to have something to compare NAT16 to and because MOGAT3 was already reported to be missing or a pseudogene in rodents. We do not have indications that the genes are connected on a transcript level. However, it is possible that there have been chromosome level events during evolutionary history that have

had an impact on multiple genes. One such event could perhaps explain why we found many fewer NAT16 and MOGAT3 genes than VGF genes in rodents.

48. Line 373, “poor quality”: Can the authors please define how they define “poor quality” here. How can sequences be of “poor quality”?

RESPONSE: The definition was present in line 410-411. Essentially, these sequences have been “modified relative to the genome sequence to correct for possible protein-altering mismatches or indels in the genome sequence”. We have changed the text to indicate the sequences were (not) labeled “low quality protein”.

49. Lines 384 and following: the presence of NMT1/2 and NAT16 in bacteria, archaea and eukarya is a bit confusing. In line 385 the authors state that NMTs have been reported to be limited to eukaryotes while later in line 393 they state that both structures are present in archaea and bacteria. Please explain this more precisely.

RESPONSE: This comment captures quite well what we attempted to convey here. We’ve read that NMTs are only present in eukaryotes. We considered that perhaps NAT16-type proteins are an ancestral form of this enzyme that led to the NMT-type during eukaryogenesis. However, here we found similar structures to NMTs are also present in bacteria and archaea, making the relationship between the two groups difficult, perhaps impossible, to understand.

We have clarified the text to make it better understandable.

50. Might this indicate that NMT1/2 and NAT16 developed in terms of divergent evolution from a common ancestor?

RESPONSE: This is what we have attempted to understand in this section. The conclusion is that it is not clear since both types of structure are present in all domains of life suggesting either that both types were present before the split of primordial life to bacterial and archaeal lineages, or that one or both types emerged either in bacteria or archaea and were transferred to the other lineage by horizontal gene transfer. See also previous comment.

51. Line 443, “was not set up to find them”: can the authors explain why these acylated histidines could not be detected. Can the authors re-evaluate the data?

RESPONSE: It is more accurate to state that these metabolites were not identified in the dataset, and we have amended the text in this regard. There are molecules in the dataset that have been identified by mass alone that match the molecular weight and formula of propionyl- and butyryl-histidine (identified as zalcitabine/dideoxycytidine and 4-(methylnitrosamino)-1-(3-pyridyl-n-oxide)-1-butanol, respectively). These molecules were increased in the cell lines overexpressing NAT16 but were not changed in WT vs. KO, suggesting they could be produced by overexpressed NAT16. We did not see similar patterns for molecules whose mass would match that of larger acylhistidines such as palmitoyl- or myristoylhistidine. We have mentioned the detected molecules in the discussion (line 496-498).

52. It is not clear how the acetyl-histidine is entering the blood? It is generated by a cytosolic enzyme. What might be the role of acetyl-histidine in cells? Can the authors speculate? Storage for free histidine? Or is it derived from peptides/proteins? Does it regulate protein function similar as described for other NATs, i.e. regulating protein stability or turnover. It could also act co-translationally if the first residue is His; or if the initiator methionine is removed, the second residue is a His. In this case it might be active towards peptides but not towards folded proteins.

RESPONSE: Based on high expression of NAT16 in endocrine cells/tissues (Supplementary figure 4), its cytosolic subcellular localization (Figure 2), and the general transport routes for amino acids across the plasma membrane, acetyl-histidine is likely to be produced in the cytosol and thereafter released into the blood. Based on our new data (Supplementary table 1) and the structural data, there are no indications that HisAT acetylates histidine in peptides or proteins.

We have discussed the potential roles of acetylhistidine, including cellular roles, in the discussion section paragraphs starting at lines 542 and 576.

53. Is these also non-enzymatic acetylation on histidines?

RESPONSE: We did see elevated background signal in the pH/buffer screen at pH >8.5 where a larger proportion of the histidine α -amino groups would be deprotonated potentially promoting this reaction. We are not aware of it occurring to a large extent at neutral conditions.

54. Do the authors have any indication if catalysis occurs by a general base mechanism. For many GNATs a catalytic Glu acts as general base. Is there a conserved Glu? Can the authors make a mutational analysis? This Glu should be in the area where the His-N-terminal amino group is located so that it can abstract a proton increasing its nucleophilicity to attack the electrophilic carbon of the acetyl group. Maybe the structures can resolve this.

RESPONSE: There are no completely conserved glutamates in the sequences shown in Supplementary Fig. 11. The mutation of the glutamate closest in the structure to the substrate histidine, Glu122 (6.7 Å from the closest O-atom to His α -N), to a glutamine slowed down the catalysis a little but did not inactivate the enzyme. Glu108 is also fairly close to the substrate (9.9 Å from the closest O-atom to His α -N), but is not completely conserved in vertebrate sequences, and its mutation to glutamine did not reduce activity. It is possible that the deprotonation for the catalysis is mainly carried out by Glu122, but there is some redundancy in the mechanism. It is also possible that Glu122 has another role during catalysis or substrate recognition and the deprotonation is carried out by another residue, possibly transmitted via water molecules. We have included new data on relevant NAT16 mutants in Supplementary Figure 5c-d.

55. Methods: Dpnl; write Dpn in italics; if Nat. Communications still sticks to the rule.

RESPONSE: The text was modified as suggested.

Reviewer #2 (Remarks to the Author): The reviewer was only tasked to review technical details of the metabolomics analyses.

The following points arise: (a) data are not uploaded to public repositories, e.g. MetaboLights or MetabolomicsWorkbench. Authors must upload both raw instrument files and result data. The authors make a false statement when they declare in the NR reporting summary that all other data are included with the manuscript. The metabolomic data has not been included in either the manuscript or the supplement. The supplement only includes compound names, RT and m/z but not MS/MS information, and not the library that was used for compound ID and MS/MS similarity thresholds. The supplement does not include the actual data, but only the statistics results. The supplement does not include the instrument raw data, either.

RESPONSE: The data are now uploaded to MetaboLights database with the id MTBLS12070 and are accessible via the following URL (<https://www.ebi.ac.uk/metabolights/reviewer6f53c866-cd81-4513-a38d-a14a46266555>).

(b) cells were extracted by methanol/chloroform (80:20), excluding highly hydrophilic compounds such as sugar acids. This is probably fine with the aim of the analyses that looked at amino acids, but should be discussed as a limitation. Given the very lipophilic extraction method, it is strange to see that the authors did not detect lipids in their 'semi-polar' chromatography method.

RESPONSE: We have included a sentence about the extraction method excluding highly hydrophilic compounds in the results section. There were some lipids detected, but these were only identified based on mass (level 3).

(c) The authors do not disclose the chromatography methods, but refer to two former publications, and even say that methods were modified ("using a slightly modified version of the protocol described by Hsiao et al. 132 for polar metabolites or by Doneanu et al. for semi-polar metabolites."). This is not acceptable reporting practice. Methods must be given in detail to allow other researchers to repeat analyses. Method details can also be given in supplements, if needed, and if method sections get too long otherwise.

RESPONSE: The detailed chromatography methods are included in the MetaboLights submission and in the methods section including the supplementary table 10.

(d) Compound annotations is likely correct, and authors give some criteria (but not RT windows and not MS/MS similarity thresholds). Software Compound Discoverer is fine, but the authors do not report the confidence levels reported by that software, and authors do not report the parameters used for data processing. A number of compounds are not fully annotated ('hexose' or 'sugar alcohol'), and it is unclear which external library was used for compound annotations.

RESPONSE: These details are now included in the MetaboLights submission and the methods section. Accepted RT deviation was 0.2 min, accepted mass deviation 3 ppm, and MS/MS match factor was >60.

Reviewer #3 (Remarks to the Author):

The paper by Myllykoski et al provides new, important information on Nat16, as the enzyme that acetylates histidine in specific cells and tissues in humans. Until now, this function had only been described in a fish, and the human recombinant enzyme had been tested, but was apparently without any activity. The function is now clear and the role of Nat16 needs to be found. Nat16 is present in many vertebrates and in more primitive organisms, but it is apparently absent from most rodents, though, as stressed by the authors, the difficulty of sequencing Nat16 in many species, makes that it is difficult to be sure about this absence. The paper provides a quite complete kinetic investigation of Nat16 and provides its structure, with its peculiar double GNAT fold, which is compared with that of other enzymes of the GNAT family.

Overall the paper provides a lot information and provides a good coverage of the literature. This paper is a major step in the elucidation of the function of N-acetylhistidine, which is predominantly expressed in brain and secretory tissues. The experiments have been carefully performed and well analysed.

I have only minor comments

Summary Line 22 ...Lower blood levels of acetylhistidine which impacts kidney function

This a strong statement: nothing is known on how a reduced acetylhistidine level could cause kidney dysfunction.

RESPONSE: We have removed this strong statement.

Figure 1. Arginine and lysine are substrates for Nat16, as shown in Fig1A. Since the results in Fig. 1a underestimate the activity on histidine, it would be good to provide values of K_m and V_{max} of the enzyme when lysine or arginine are used as a substrate. There is no need to perform as detailed a kinetic investigation as performed for histidine.

RESPONSE: We have carried out enzyme assays with varied arginine and lysine concentrations at fixed Ac-CoA concentration. The results are included in Supplementary Figure 2h-i and indicate that both are on a similar level as substrates whereas the catalytic rate is slightly lower and K_M slightly higher for lysine. Both have more than one order of magnitude difference to histidine in both k_{cat} and K_M . Together with the metabolomics data, these results suggest that arginine and lysine are probably not substrates for NAT16 except in exceptional situations, like when NAT16 is artificially overexpressed.

Line 117. Please provide information on the concentration of histidine used for determining the kinetic constant in Fig. 1C

RESPONSE: We have elaborated on details of the enzyme assays in the Figure 1 legend.

Figure 2C : overexpression of Nat16 seems to cause higher increases in acetyllysine than in N-acetylarginine, yet arginine seems to be a better substrate on the purified enzyme than lysine. What does 'relative abundance' mean ? Are the values for the different metabolites comparable ?

RESPONSE: The relative abundance is derived from the area under the curve of the chromatography peak. These values are not comparable between different metabolites, so

we cannot use these results to compare which is better *in vivo* substrate, arginine or lysine, or to determine how much of the cellular histidine pool is acetylated. We have clarified this in the figure legend.

Line 609, page 25; were the kinetic studies performed with a whole set of Acetyl-CoA concentrations combined with a whole set of histidine concentration, as the sentence suggests ?

RESPONSE: Yes. For the first two series we used slightly staggered setup where the histidine concentrations for the three lowest Ac-CoA concentrations were slightly different than for the highest three (0, 20, 50, 120, 300, 720, 1800, or 4500 μM histidine for lower Ac-CoA concentrations in Series 1 and 2 and all Ac-CoA concentrations in Series 3, and 0, 50, 120, 300, 720, 1800, 4500, or 10000 μM histidine for higher Ac-CoA concentrations in Series 1 and 2). We have clarified these in the method section.

Supplementary figure 2, legend; please indicate the concentration of histidine that was used

RESPONSE: We have included the concentrations in the legend.

**Review to the manuscript entitled “*The molecular basis for acetyl-histidine synthesis by HisAT/NAT16*”**
**submitted by Myllykoski and co-authors**

The manuscript entitled “*The molecular basis for acetyl-histidine synthesis by HisAT/NAT16*” submitted by
Myllykoski and co-authors describes the structural and functional characterization of the GNAT-acyltransferase
NAT16 as N-(α)-histidine acyltransferase. The authors performed enzyme kinetics applying an DTNB assay to
detect the thiol groups of released coenzyme A to study if NAT16 uses proteinogenic amino acids and further
substrate candidates as substrates. These pre-screening studies indicated that NAT16 is capable to use histidine
as major substrate but that it is also capable to use arginine, lysine and the aromatic amino acids phenylalanine
and tyrosine as substrates. The authors confirmed these *in vitro* studies by performing metabolomics studies from
haploid HAP1 cells generating a $\Delta nat16$ knockout cell line to show that the knockout has an impact on the acetyl-
histidine level while not affecting the acetylation level of other amino acids. The authors show that the enzyme
NAT16 is localized in the cytosol and conclude from a public database that the expression is high in retina, kidney
and brain. Moreover, the authors conduct Michaelis-Menten kinetics and propose these data suggest an enzymatic
mechanism involving formation of a ternary complex between enzyme, substrate, i.e. histidine, and co-substrate,
i.e. acetyl-CoA, and in which an tetrahedral intermediate is formed. The authors characterize a mutation of NAT16,
i.e. F63S, which was shown to result in an decrease in blood serum acetyl-histidine, which correlates with certain
forms of kidney disease. Biochemical characterization show this mutant shows a slightly increased K_M value
suggesting that it interferes with binding affinity towards histidine while not affecting catalysis. Finally, the authors
present several crystal structures of NAT16 in complexes with histidine and CoA-SS-CoA disulfide, arginine and
CoA-SS-CoA disulfide, imidazole and CoA-SS-CoA disulfide, histidine and ethyl-CoA, and myristoyl-histidine and
CoA. The authors provide a characterization of the substrate binding site, suggest potential residues involved in
formation of the oxyanion hole during catalysis but do not present data on a potential general base. The authors
show that NAT16 is structurally related to N-terminal myristoyl-transferase 1/2 (NMT1/2) as both have a double
GNAT-domain architecture. The phylogenetic data/dendrogram should show some evolutionary development of
NAT16 and NMT1/2 but is a bit confusing. Overall, the physiological significance of the enzyme NAT16 is not
resolved and the role of acetylation of histidine side chains is unclear. In total the manuscript is important and gives
novel insights into acetyltransferases with direct correlation to human diseases. To this end, I recommend in
principle the publication of the manuscript but certain issues need to be resolved prior to publication as explained
in the next section.

**Open Points:**

- 1. There is a class of bacterial GNAT enzymes, i.e. type V GNATs, which also encompass a double-GNAT domain
which is followed by a C-terminal domain. Does NAT16 show similarities, i.e. in terms of biochemistry, structure
and function, to these type V GNATs?
- 2. Avoid emotional or judging language such as “overwhelmingly” in line 91, “good” in line 92, “worse” in line 257.
Describe it scientifically. What do you precisely want to say with “good”, what do you define as “worse”, etc.
- 3. Line 95: write min instead of minutes and add the unit also to the number 30.
- 4. It is not clear from the manuscript if NAT16 has any peptide or protein substrates. The authors describe that
they tested various peptides which were shown in a table in the SI but these peptides do not contain an N-
terminal His. The authors should test if NAT16 is capable to acetylate peptides or proteins with an N-terminal
His. This would have further implications for the function of the enzyme modulating peptide/protein function and
processes such as protein stability/turnover. Along that line could the increase in blood acetyl-lysine also be
due to degraded proteins that carried an acetyl-histidine at the N-terminus rather than due to acetylation of
isolated histidine side chains?
- 5. The authors performed metabolomics to show that knockout of NAT16 resulted in a decrease in acetyl-lysine
levels. Furthermore, they state that for lysine it is not clear if lysine is acetylated at the side chain, i.e. N-(ϵ)-
acetylation, or at the N-terminus, i.e. N-(α)-acetylation. Can the authors explain why they can be sure that
histidine is not acetylated at the δ - or ϵ -N of the imidazole ring? If it is in its unprotonated state it might be
competent to attack as a nucleophile the electrophilic carbonyl carbon of the acetyl-group of acetyl-CoA.
Histidine side chains can also be phosphorylated and these side chains nitrogens. Can the authors show this
without any doubt, i.e. does the MS/MS measurement result in fragmentation spectra.
- 6. The authors show binding of a oxidized double-CoA molecule, i.e. two CoA molecules linked together covalently
by forming a disulfide bond via their thiol groups. Do the authors think that both binding sites are important for
the activity of the enzyme? Are both saturated in their enzyme kinetics? How are the affinities for both sites? It
might play an important regulatory role if for example the second CoA-binding site has a lower affinity compared

- to the catalytic acetyl-CoA binding suggesting that it might impair enzyme activity at very high CoA-
concentrations. Along that line, would it be competent structurally to accommodate two acetyl-CoA molecules
or is binding of the CoA-SS-CoA only possible with two CoA molecules or is it competent to bind one acetyl-
CoA and one CoA? Along that line, the authors say that they observe significant substrate inhibition at higher
acetyl-CoA concentrations. I do not see the substrate inhibition in the data (Supp. Fig. 2). Where is it shown?
Can the mechanism underlying the mode of substrate inhibition have sth. To do with saturating bot CoA binding
sites? Maybe there is some CoA in the acetyl-CoA stock, it is quite unstable.
- 7. Fig. 1 A: please show mean values and standard deviations (and not just their “average”) and write in the figure
legends how often you repeated the experiments and which statistics tests were performed. How did you
determine the CoA-concentrations? Did you generate a calibration curve? Can you show it in the SI section.
 - 8. Fig. 1B: can you not draw chemical structures yourself instead of adapting it from a different source? Why do
you show different His tautomers following acetylation? Is the tautomer protonated at N- δ favored if histidine is
acetylated?
 - 9. Fig. 1C: Can the authors explain where these numbers come from? Maybe show here the mean values and
standard deviations. Also show the k_{cat}/K_M values as this is the value that should be used to compare the
enzymes’ efficiencies.
 - 10. Supp. Fig. 2: The authors show the Michaelis-Menten kinetics. In this case the enzyme uses two (co)substrates,
i.e. histidine and acetyl-CoA. In panel A the authors show three series of MM-kinetics. Can the authors not also
show the values for the higher acetyl-CoA concentrations to see the effect of substrate inhibition? The authors
could fit the data with a model “Michaelis-Menten kinetics with substrate inhibition” which would also yield the
K_i for the substrate inhibition:
$$V_0/[E]_0 = k_{cat}[S]/(K_M+[S]/(1+[S]/K_i))$$
 - 11. The authors derive from the data shown in Supp. Fig. 2B that the enzyme uses a sequential mechanism, i.e.
formation of a ternary complex, and furthermore that formation of a tetrahedral intermediate is obvious. If you
have an enzyme with two substrates it is correct to perform MM-kinetics varying both concentrations as done
here. Panel A suggests that all MM-kinetics will result in the same (or highly similar K_M -value for [His]). However,
in Fig. B there is no Lineweaver Burk plot shown, which would show $1/V_0$ at the y-axis and $1/[S]$ on the x-axis,
but this is not the case. If this would be the case and the lines would cross the x-axis (and not the y-axis as
written in the figure legend!) in the same point, it would indicate a sequential kinetics. I do not get the point why
the authors concluded from these data that the mechanism proceeds via a tetrahedral intermediate.
 - 12. For GNATs reports are available that confirm and support the sequential kinetics mechanism. These should be
cited.
 - 13. MM-kinetics: it would have been straightforward to keep one concentration constant (at saturating level) and
just change the other concentration to determine K_M and k_{cat} for one (co)substrate and do it vice versa for the
other substrate.
 - 14. Supp Fig. 1C/D: Please explain exactly what is plotted here and where the data were obtained from that were
used for these plots.
 - 15. Fig. 1C: The K_M for His can be obtained from the MM-kinetics plots shown in Supp. Fig. 2A. Can the authors
explain how they obtained the K_M values for acetyl-CoA and how they determined the k_{cat} values shown in 1C.
 - 16. Supp. 1D: which data were derived from Supp. Fig. 2D. Please explain to be able to follow it.
 - 17. MM-kinetics: show also the primary data, i.e. [CoA] as a function of time.
 - 18. Line 138, “...assay did not reveal the location of the acetyl group”: please explain better what you want to say
here, i.e. either acetylation at N-(α)- or N-(ϵ) of lysine side chain.
 - 19. Line 132, “...was significantly decreased”: did the authors perform a statistics test? If yes show it and say in the
legend what test was conducted and show the significance level.
 - 20. Fig. 2: Write “Western” with capital “W” as the name is derived from a surname.
 - 21. Fig. 2C: Can the authors judge how the stoichiometry of acetylation on histidine is. As shown by the
metabolomics, they identified histidine and acetyl-histidine in the cells. Were both present at similar
amount/concentration? Can the authors conclude anything concerning the stoichiometry?
 - 22. Line 165, “This agrees with a model in which”: can the authors explain which model they refer to and cite the
reference.
 - 23. Lines 174/175: Please show a clear image of the structure from which the location of the mutation F63S
becomes clear. Is it near the CoA binding site? Is it near the His binding site?
 - 24. Lines 177-180: the authors show that the mutation F63S in NAT16 affects the K_M value and only slightly k_{cat} .
They state that an effect is observable “especially at low histidine conditions”. It is important to note, that the
concentrations matter, so replace “low histidine conditions” with “low concentrations of histidine”. If the enzyme

is saturated with histidine no effect of the mutant is detectable as k_{cat} is almost unchanged. In this respect, is it
known how the intracellular concentrations of histidine are in the tissues where NAT16 is expressed. Is a
difference of the activity of the mutant compared to wildtype expectable? This will help to judge the physiological
importance of the mutant. This could be discussed.

25. Fig. 3A,B: this is a figure derived from a public database. I suggest moving it into the SI section.

26. If seeing Fig. 3A/B the question is if the differences in the mRNA levels are manifested on the protein level in
the tissues.

27. Fig. 3C: the structural models are not really helpful to see where the mutation is located. Is it an AlphaFold
model or which structure is used here? The only difference in the two subpanels is the color. I do not see a
clear benefit showing this.

28. Fig. 3C: the bar graphs lack correct labels on the y-axis, i.e. label and unit.

29. Check throughout if all graphs have correct labels on the axes; there are graphs in the main section and in the
SI section that lack properly labelled axes.

30. Just for curiosity, can the author explain why they used isomorphous replacement to solve the phase problem
and not molecular replacement? AlphaFold2 generates excellent molecular replacement models or did it not
work out for the authors?

31. Line 202: "coenzyme A disulfide molecule": the authors should explain a bit better, what they want to say. i.e. it
is two CoA covalently connected by formation of a disulfide bridge with their thiols.

32. Line 203: "crystallized also in space group I222": why do the authors say "also" here; it is the first time they
mention this space group in the manuscript. That is a bit confusing.

33. Line 204: "the truncated N-terminus interacted", Line 206: "whole truncated N-terminus was visible": these
statements are really confusion. How can a part that is truncated, i.e. not present, interact with sth. or can be
visible. Please work on the language and explain it differently.

34. Show a 2Fo-Fc map for all ligands.

35. Show a proper Fo-Fc omit map for all ligands and replace the polder map with those. A polder map is not a real
omit map.

36. All structure images: the images should be labelled better. Label all important parts, side chains, CoA, α -
helices, β -strands, shown. Some figures are really crowded. Please work on the images to make it better
readable.

37. The ethyl-CoA is firstly mentioned in Fig. 4 and line 241 but it is explained later in lines 266 and following. Please
shift this section to explain the compound when it is introduced.

38. Give information on how the structures with ligands were obtained, was it by soaking or by co-crystallization?

39. Line 261: The authors state that differences in the affinities might explain the difference observed in enzyme
activity between arginine and histidine and poor activity towards other amino acids. If so, the authors should
observe activity under saturating concentrations of the amino acids.

40. Line 257, "lower resolution cutoff": the cutoff is set by the user, applying certain rules where to set the cutoff for
the resolution, i.e. $CC_{1/2}$ (or $I/\sigma I$, R_{meas}/R_{merge}).

41. Line 258, "The cause for the HisAT substrate preference...". The authors explained that the imidazole ring can
form quite some interactions with side chains and main chains in the enzyme's active site. Why does not explain
the specificity. Aromatic side chains, Phe and Tyr, might form the stacking interactions with the nearby aromatic
side chains in the active site, and Trp might just be too bulky to fit into the active site, i.e. it would sterically not
be possible to be a substrate.

42. Line 271, "These two amides generate an oxyanion hole...". Is that speculation or is that derived from structural
or sequence alignments? Are these residues conserved in other GNATs? Do the authors show this in a figure?

43. For the section of comparison of NAT16 with NMT1/" show a structural superposition and calculate R.M.S.D.
values to get a value for their similarity.

44. Line 304: Arg161 and Asp163 are not visible in the electron density of the substrate structure. Are they in flexible
parts which are fixed upon myristoyl-histidine binding?

45. Line 318: "...that clearly contributes to enzymatic activity or substrate binding is Trp246". How do the authors
conclude this? Is that based on the structures showing that it is structurally involved binding the histidine? Do
the authors have MM-kinetics for the mutant?

46. Line 326, "... and myristoylated peptide substrate in NMT1": if the binding of substrate so similar in both
enzymes, maybe this is a hint for NAT16 also being able to act as peptide/protein N-(α)-acetyltransferase if a
histidine is at the N-terminus and not only active on the isolated amino acid similar to other N-terminal
acetyltransferases acting co- or posttranslationally. Maybe the efficiency is higher for a peptide/protein as other
parts apart from the histidine side chain contribute to binding affinity.

- 47. Lines 349 and following: Can the authors explain why the neighboring genes to *nat16* are so important? Do
the authors have indications that these are somehow connected on the transcriptional or post-transcriptional
level?
- 48. Line 373, "poor quality": Can the authors please define how they define "poor quality" here. How can sequences
be of "poor quality"?
- 49. Lines 384 and following: the presence of NMT1/2 and NAT16 in bacteria, archaea and eukarya is a bit confusing.
In line 385 the authors state that NMTs have been reported to be limited to eukaryotes while later in line 393
they state that both structures are present in archaea and bacteria. Please explain this more precisely.
- 50. Might this indicate that NMT1/2 and NAT16 developed in terms of divergent evolution from a common ancestor?
- 51. Line 443, "was not set up to find them": can the authors explain why these acylated histidines could not be
detected. Can the authors re-evaluate the data?
- 52. It is not clear how the acetyl-histidine is entering the blood? It is generated by a cytosolic enzyme. What might
be the role of acetyl-histidine in cells? Can the authors speculate? Storage for free histidine? Or is it derived
from peptides/proteins? Does it regulate protein function similar as described for other NATs, i.e. regulating
protein stability or turnover. It could also act co-translationally if the first residue is His; or if the initiator
methionine is removed, the second residue is a His. In this case it might be active towards peptides but not
towards folded proteins.
- 53. Is these also non-enzymatic acetylation on histidines?
- 54. Do the authors have any indication if catalysis occurs by a general base mechanism. For many GNATs a
catalytic Glu acts as general base. Is there a conserved Glu? Can the authors make a mutational analysis? This
Glu should be in the area where the His-N-terminal amino group is located so that it can abstract a proton
increasing its nucleophilicity to attack the electrophilic carbon of the acetyl group. Maybe the structures can
resolve this.
- 55. Methods: Dpnl; write Dpn in italics; if Nat. Communications still sticks to the rule.

Response to the authors' comments to the manuscript entitled "*The molecular basis for acetyl-histidine synthesis by HisAT/NAT16*" submitted by Myllykoski and co-authors.

The manuscript did strongly improve in this round of revision. I do not have any further open points which need to be addressed by the authors as you see on my comments (in red) to the answers supplied by the authors. Overall, it is a highly interesting study and I recommend publication of this manuscript in its current form.

Rebuttal letter for "*The molecular basis for acetyl-histidine synthesis by HisAT/NAT16*".

We thank the Editor and reviewers for their useful inputs to our work and for the invitation to resubmit an improved manuscript version. We have carried out additional experiments and adjusted the manuscript text and figures according to reviewer comments, and we hope that the revised manuscript is found acceptable for publication. Please see specific actions and responses to reviewer comments below.

Reviewer #1 (Remarks to the Author):

Review to the manuscript entitled "*The molecular basis for acetyl-histidine synthesis 1 by HisAT/NAT16*" submitted by Myllykoski and co-authors

The manuscript entitled "*The molecular basis for acetyl-histidine synthesis by HisAT/NAT16*" submitted by Myllykoski and co-authors describes the structural and functional characterization of the GNAT-acyltransferase NAT16 as N-(a)-histidine acyltransferase. The authors performed enzyme kinetics applying an DTNB assay to detect the thiol groups of released coenzyme A to study if NAT16 uses proteinogenic amino acids and further substrate candidates as substrates. These pre-screening studies indicated that NAT16 is capable to use histidine as major substrate but that it is also capable to use arginine, lysine and the aromatic amino acids phenylalanine and tyrosine as substrates. The authors confirmed these *in vitro* studies by performing metabolomics studies from haploid HAP1 cells generating a *Dnat16* knockout cell line to show that the knockout has an impact on the acetylhistidine level while not affecting the acetylation level of other amino acids. The authors show that the enzyme NAT16 is localized in the cytosol and conclude from a public database that the expression is high in retina, kidney and brain. Moreover, the authors conduct Michaelis-Menten kinetics and propose these data suggest an enzymatic mechanism involving formation of a ternary complex between enzyme, substrate, i.e. histidine, and co-substrate, i.e. acetyl-CoA, and in which an tetrahedral intermediate is formed. The authors characterize a mutation of NAT16, i.e. F63S, which was shown to result in an decrease in blood serum acetyl-histidine, which correlates with certain forms of kidney disease. Biochemical characterization show this mutant shows a slightly increased KM value suggesting that it interferes with binding affinity towards histidine while not affecting catalysis. Finally, the authors present several crystal structures of NAT16 in complexes with histidine and CoA-SS-CoA disulfide, arginine and CoA-SS-CoA disulfide, imidazole and CoA-SS-CoA disulfide, histidine and ethyl-CoA, and myristoyl-histidine and CoA. The authors provide a characterization of the substrate binding site, suggest potential residues involved in formation of the oxyanion hole during catalysis but do not present data on a potential general base. The authors show that NAT16 is structurally related to N-terminal myristoyl-transferase 1/2 (NMT1/2) as both have a double GNAT-domain architecture. The phylogenetic data/dendrogram

should show some evolutionary development of NAT16 and NMT1/2 but is a bit confusing. Overall, the physiological significance of the enzyme NAT16 is not resolved and the role of acetylation of histidine side chains is unclear. In total the manuscript is important and gives novel insights into acetyltransferases with direct correlation to human diseases. To this end, I recommend in principle the publication of the manuscript but certain issues need to be resolved prior to publication as explained in the next section.

Open Points:

1. There is a class of bacterial GNAT enzymes, i.e. type V GNATs, which also encompass a double-GNAT domain which is followed by a C-terminal domain. Does NAT16 show similarities, i.e. in terms of biochemistry, structure and function, to these type V GNATs?

RESPONSE: Type V GNATs are a group of bacterial enzymes of which the *Mycobacterium tuberculosis* enzyme Eis (Enhanced intracellular survival) seems to be the best characterized. Eis is a secreted enzyme that acetylates a broad range of substrates including aminoglycosides, lysine side chains of peptides, and arylalkylamines such as histamine. Eis type enzymes consist of two GNAT domains arranged in a similar way to NAT16/HisAT and an additional C-terminal domain with homology towards sterol carrier proteins (Chen et al. 2011, PMID: 21628583).

Structurally Eis and NAT16/HisAT are similar in that they both have the double-GNAT fold and for both the N-terminal of the two GNAT domains is the active one. They are different in that Eis has the additional C-terminal domain, and that NAT16/HisAT appears to be a monomer while Eis has a conserved hexameric tertiary structure. Eis has also a more open substrate binding cavity between the GNAT domains, possibly reflecting the broader substrate range. The carboxyl group of the very C-terminal residue of Eis enters the active site and is involved in catalysis in the same way as with NMTs but unlike with NAT16/HisAT, and Eis does not have a corresponding β -hairpin that we used to identify NAT16/HisAT type structures, suggesting Eis structure is closer to the NMT-type than the HisAT-type.

Eis has broad activity towards different aminoglycosides often acetylating multiple amides of one molecule, lysine side chains of peptides, and even arylalkylamines such as histamine (Chen et al. 2011, PMID: 21628583, Kim et al. 2012, PMID: 22547814, Pan et al. 2018, PMID: 29402941), while NAT16/HisAT appears to be more focused with activity towards histidine and residual activity towards other amino acids. Eis enzymes may also be more limited in their ability to accept longer acyl-CoA substrates (Green et al. 2015, PMID: 27622743). As far as we can tell, it is not known if Eis enzymes can acetylate histidine, but given their broad substrate specificity, and the capacity to acetylate histamine, it would not be very surprising.

Functionally, the secreted *M. tuberculosis* Eis is thought to confer resistance against aminoglycoside antibiotics such as kanamycin and other anti-TB drugs by acetylating them, and to enhance the survival of the bacteria within macrophages by the activation of DUSP16 via its acetylation at lysine 55. NAT16/HisAT is an intracellular cytosolic histidine acetyltransferase, but the roles of this activity and its product are currently unknown.

We initially overlooked this group of enzymes, as they didn't score highly in similarity searches possibly because of the extra domain. We have now mentioned them in the introduction (line 74) and discussion (line 518). In addition, we repeated the Dali structural similarity analysis with two mycobacterial Eis proteins included in a revised Figure 6a. The placement of the Eis proteins in the resulting dendrogram as early branches of the NMT-/HLS-type half of the dendrogram appears to confirm our suggestion above that they are structurally closer to NMT-type than HisAT-type.

Response:

The authors concisely answered this open point.

2. Avoid emotional or judging language such as “overwhelmingly” in line 91, “good” in line 92, “worse” in line 257. Describe it scientifically. What do you precisely want to say with “good”, what do you define as “worse”, etc.

RESPONSE: We have removed or rephrased these words.

Response:

The authors answered this open point.

3. Line 95: write min instead of minutes and add the unit also to the number 30.

RESPONSE: We modified the text as suggested. (Line 96)

Response:

The authors answered this open point.

4. It is not clear from the manuscript if NAT16 has any peptide or protein substrates. The authors describe that they tested various peptides which were shown in a table in the SI but these peptides do not contain an N terminal His. The authors should test if NAT16 is capable to acetylate peptides or proteins with an N-terminal His. This would have further implications for the function of the enzyme modulating peptide/protein function and processes such as protein stability/turnover. Along that line could the increase in blood acetyl-lysine also be due to degraded proteins that carried an acetyl-histidine at the N-terminus rather than due to acetylation of isolated histidine side chains?

RESPONSE: As far as we know, NAT16 does not have peptide or protein substrates. We have now tested selected His-starting peptides (together with Ser-starting controls) that have been reported to exist in cells and some of which were reported to be acetylated (Yeom et al. 2017, PMID: 28747677). There was no detectable activity towards these peptides, see revised Supplementary Table 1 and comment in the Results, (line 100). Considering the clear acetyltransferase activity towards histidine and no activity towards peptides with histidine or other termini, we do not believe the proposed mechanism is a likely source of acetylhistidine.

Response:

The authors answered this open point to my complete satisfaction.

5. The authors performed metabolomics to show that knockout of NAT16 resulted in a decrease in acetyl-lysine levels. Furthermore, they state that for lysine it is not clear if lysine is acetylated at the side chain, i.e. N-(e)-acetylation, or at the N-terminus, i.e. N-(a)-acetylation. Can the authors explain why they can be sure that histidine is not acetylated at the d- or e-N of the imidazole ring? If it is in its unprotonated state it might be competent to attack as a nucleophile the electrophilic carbonyl carbon of the acetyl-group of acetyl-CoA. Histidine side chains can also be phosphorylated and these side chain nitrogens. Can the authors show this without any doubt, i.e. does the MS/MS measurement result in fragmentation spectra.

RESPONSE: There was no difference in acetyllysine levels in NAT16 knockout vs. WT cells (without NAT16 overexpression).

The metabolomic analysis identified acetylhistidine at level 1 based on three criteria: identification by retention times (compared against in-house authentic standards +/-0.2min), accurate mass (with an accepted deviation of 3ppm), and MS/MS spectra (match factor threshold >60). The first of these was missing for the acetyllysine metabolite, making its reliable identification less certain. There are also multiple additional lines of evidence pointing towards N-alpha acetylation. A methylated nitrogen would be expected to block the acetylation of the same nitrogen in the histidine side chain, but both methylhistidines were acetylated similarly to the unmethylated histidine. In the crystal structure the alpha-amino group of histidine is positioned right next to where the acetyl group of Ac-CoA would be, while the side chain nitrogens are further away. Finally, there are numerous reports in the literature reporting N-alpha-acetylhistidine being detected in animal tissues. The reports of side chain acetylated histidines are very rare.

Response:

The authors convincingly answered this open point.

6. The authors show binding of a oxidized double-CoA molecule, i.e. two CoA molecules linked together covalently by forming a disulfide bond via their thiol groups. Do the authors think that both binding sites are important for the activity of the enzyme? Are both saturated in their enzyme kinetics? How are the affinities for both sites? It might play an important regulatory role if for example the second CoA-binding site has a lower affinity compared to the catalytic acetyl-CoA binding suggesting that it might impair enzyme activity at very high CoA concentrations. Along that line, would it be competent structurally to accommodate two acetyl-CoA molecules or is binding of the CoA-SS-CoA only possible with two CoA molecules or is it competent to bind one acetyl-CoA and one CoA? Along that line, the authors say that they observe significant substrate inhibition at higher acetyl-CoA concentrations. I do not see the substrate inhibition in the data (Supp. Fig. 2). Where is it shown? Can the mechanism underlying the mode of substrate inhibition have sth. To do with saturating bot CoA binding sites? Maybe there is some CoA in the acetyl-CoA stock, it is quite unstable.

RESPONSE: We think that only the canonical Ac-CoA binding site is required for catalysis. We have not measured the affinities of the different binding sites. It is not clear to what extent the second site is

(Ac-)CoA binding site and to what extent it is an artefact of crystallization. We only managed to produce the P63 crystals when both CoA molecules were present, and the second one was generating a crystal contact. Further, it may be that the C-terminal 6xHis tag that was part of this crystal contact, is also required for its binding. There was no electron density in the second CoA binding site in the structure with S-ethyl-CoA even though the concentration for this substrate analog was 0.7 mM in the crystallization buffer and 1 mM in the cryoprotectant soaking solution, suggesting that two Ac-CoA molecules probably cannot bind simultaneously.

Response:

The authors concisely answered this open point.

We identified the substrate inhibition from the decrease in the limiting rate (V_{max}) in the Michaelis-Menten curves in Supplementary Figure 2a and the values listed in Supplementary Figure 2d, with higher Ac-CoA concentrations. For example, in series 2 and 3, the reactions with 500 μ M Ac-CoA have similar limiting rates as the reactions with 12 μ M Ac-CoA, while the highest rates are seen with reactions containing 30 μ M Ac-CoA. Please see also the response to comment 10 and the new Supplementary Figure 2e.

The mechanism of substrate inhibition could potentially be the result of Ac-CoA or CoA binding to the second site preventing productive binding to the canonical site, but there is no solid evidence for the inhibitory mechanism at present. Cellular Ac-CoA levels are typically reported to range from low micromolar to few tens of micromolar. At those Ac-CoA levels the inhibitory properties may be irrelevant. We assumed that any CoA derived from the Ac-CoA stock present in the reaction mixture before enzyme addition would react with the DTNB present and not be available for product inhibition.

Response:

The authors answered this open point.

7. Fig. 1 A: please show mean values and standard deviations (and not just their “average”) and write in the figure legends how often you repeated the experiments and which statistics tests were performed. How did you determine the CoA-concentrations? Did you generate a calibration curve? Can you show it in the SI section.

RESPONSE: We have modified Figure 1a to show the individual data points, their mean, and the standard deviation as instructed in this comment and the editorial guideline. We have modified the legend as instructed. The CoA concentrations were determined using Beer’s law using the pathlength correction described in the reference 107 and the extinction coefficient $13700 \text{ M}^{-1} \text{ cm}^{-1}$. In this pathlength correction procedure, absorbance at 900 nm is subtracted from absorbance at 975 nm and the result is divided by a constant (0.186) to obtain the pathlength in cm.

Response:

The authors answered this open point.

8. Fig. 1B: can you not draw chemical structures yourself instead of adapting it from a different source? Why do you show different His tautomers following acetylation? Is the tautomer protonated at N-d favored if histidine is acetylated?

RESPONSE: We have drawn the structures as instructed in a revised Figure 1b. We do not think the side chain tautomer is relevant for the acetylation.

Response:

The authors answered this open point.

9. Fig. 1C: Can the authors explain where these numbers come from? Maybe show here the mean values and standard deviations. Also show the k_{cat}/K_M values as this is the value that should be used to compare the enzymes' efficiencies.

RESPONSE: We have modified Figure 1c as suggested. The K_M for Ac-CoA was derived from the negative inverse of the X-intercept in Supplementary Figure 2d. The same plot was used to derive V_{max} (not shown here) from the inverse of the Y-intercept. The Y-intercept of the plot in Supplementary Figure 2e corresponds to the K_M for histidine divided by V_{max} , so the K_M for histidine was obtained by multiplying this Y-intercept with V_{max} . The k_{cat} values were calculated from the V_{max} values by dividing with the enzyme concentration (0.02 μ M). The k_{cat} / K_M values were calculated from the k_{cat} and K_M values.

Response:

The authors answered this open point.

10. Supp. Fig. 2: The authors show the Michaelis-Menten kinetics. In this case the enzyme uses two (co)substrates, i.e. histidine and acetyl-CoA. In panel A the authors show three series of MM-kinetics. Can the authors not also show the values for the higher acetyl-CoA concentrations to see the effect of substrate inhibition? The authors could fit the data with a model "Michaelis-Menten kinetics with substrate inhibition" which would also yield the K_i for the substrate inhibition:

$$v_0/[E]_0 = k_{cat}[S]/(K_M+[S]/(1+[S]/K_i))$$

RESPONSE: The data points with Ac-CoA up to 500 μ M, which is the highest concentration we used, are already included in Figure 2a, and there is a clear decrease in the limiting rate at higher Ac-CoA concentrations. We have included the $v_0 = V_{max} [Ac-CoA] / (K_M + [Ac-CoA] (1+[Ac-CoA] / K_i))$ substrate inhibition plot for the Series 3 in Supplementary Figure 2e. The mean of the K_i from the curves at different histidine concentrations was 333 μ M (St.Dev. 19 μ M).

Response:

The authors answered this open point.

11. The authors derive from the data shown in Supp. Fig. 2B that the enzyme uses a sequential mechanism, i.e. formation of a ternary complex, and furthermore that formation of a tetrahedral intermediate is obvious. If you have an enzyme with two substrates it is correct to perform MM-kinetics varying both concentrations as done here. Panel A suggests that all MM-kinetics will result in the same (or highly similar K_M -value for [His]). However, in Fig. B there is no Lineweaver Burk plot shown, which would show $1/V_0$ at the y-axis and $1/[S]$ on the x-axis, but this is not the case. If this would be the case and the lines would cross the x-axis (and not the y-axis as written in the figure legend!) in the same point, it would indicate a sequential kinetics. I do not get the point why the authors concluded from these data that the mechanism proceeds via a tetrahedral intermediate.

RESPONSE: As is correctly stated here, the plot in panel B is not a double-reciprocal (Lineweaver-Burk) plot, but an a/v vs. a plot, sometimes called Hanes plot or Hanes-Woolf plot. Its capability to differentiate between ternary complex and substituted enzyme mechanisms is derived from the same source as it is for the double-reciprocal plot, namely the presence of the inhibitory term K_{iA} in the denominator of the rate equation for the ternary complex mechanism but not for the substituted enzyme mechanism. This leads to the easily discernible behavior of the two mechanisms when plotted on the a/v vs. a plot: In a plot for ternary complex mechanism the lines representing data points measured at different concentrations of substrate B will intercept each other at the X-axis value of $-K_{iA}$, while in a plot for the substituted enzyme mechanism they will intercept at the Y-axis ($X=0$), because the term K_{iA} is not present in the rate equation. Thus, as the lines of the plot on panel B intercept to the left of the Y-axis (at a negative value) and not on the Y-axis, we concluded that the enzyme utilizes a ternary complex mechanism.

On a double-reciprocal plot, a substituted enzyme mechanism produces lines that are parallel, while the lines representing data points measured at different concentrations of substrate B produced by a ternary complex mechanism are not parallel. A double-reciprocal plot using the same underlying data as on Supplementary Figure 2b (and the first panel of Supplementary Figure 2a) is now included as Supplementary Figure 2c. This plot also indicates a ternary complex mechanism.

Response:

The authors answered this open point.

12. For GNATs reports are available that confirm and support the sequential kinetics mechanism. These should be cited.

RESPONSE: We have discussed the GNAT family enzyme mechanisms in the discussion section (lines 514-516).

Response:

The authors answered this open point.

13. MM-kinetics: it would have been straightforward to keep one concentration constant (at saturating level) and just change the other concentration to determine K_M and k_{cat} for one (co)substrate and do it vice versa for the other substrate.

RESPONSE: Using this approach there would be no information about the kinetic mechanism. Furthermore, using saturating concentration becomes complicated in the presence of substrate inhibition.

Response:

The authors answered this open point.

14. Supp Fig. 1C/D: Please explain exactly what is plotted here and where the data were obtained from that were used for these plots.

RESPONSE: Supplementary Figure 1c is a picture of the Coomassie-stained SDS-PAGE run with the fractions from the size exclusion chromatography depicted in Supplementary Figure 1b.

Supplementary Figure 1d plots the time course of NAT16 DTNB assay to determine how rapidly NAT16 consumes all Ac-CoA in the reaction mixture of the screening assay. The plot consists of concentrations of CoA, from which a control tube signal was subtracted, as a function of time. CoA concentration was determined as described in the methods and the underlying data are included in the source data file.

Response:

The authors answered this open point.

15. Fig. 1C: The K_M for His can be obtained from the MM-kinetics plots shown in Supp. Fig. 2A. Can the authors explain how they obtained the K_M values for acetyl-CoA and how they determined the k_{cat} values shown in 1C.

RESPONSE: The K_M and V_{max} values, derived from the fitting of the Michaelis-Menten plot in Supplementary Figure 2a at fixed Ac-CoA concentrations, were considered the apparent K_M and V_{max} values. As described in Cornish-Bowden (2013, Fundamentals of Enzyme Kinetics, Wiley-Blackwell), the apparent V_{max} (V_{max}^{App}) determined by varying substrate A (histidine) at different fixed substrate B (Ac-CoA) concentrations, can be described by the equation $V_{max}^{App} = (V_{max} [Ac-CoA]) / (K_M^{Ac-CoA} + [Ac-CoA])$. As this equation is of the form of the Michaelis-Menten equation, V_{max} and K_M^{Ac-CoA} can be determined from the double-reciprocal secondary plot of $1 / V_{max}^{App}$ vs. $1 / [Ac-CoA]$ generated using the apparent V_{max} values derived from the fits of Supplementary Figure 2a and listed in Supplementary Figure 2d. This plot is shown in supplementary figure 2f (it was panel 2c in the initial submission). V_{max} was derived from the inverse of the Y-axis intercept and K_M^{Ac-CoA} from the negative inverse of the X-axis intercept. The k_{cat} values were calculated from the V_{max} values by dividing with the enzyme concentration.

We have clarified the procedure in the methods (line 692-708) and in the Supplementary figure 2 legend.

Response:

The authors answered this open point.

16. Supp. 1D: which data were derived from Supp. Fig. 2D. Please explain to be able to follow it.

RESPONSE: Supplementary Figure 2d in the initial submission is the panel 2g in current version of the manuscript.

Supplementary Figure 2g was used to determine K_M for histidine. As described in Cornish-Bowden (2013, Fundamentals of Enzyme Kinetics, Wiley-Blackwell), the ratio of the apparent constants $V_{max}^{App} / K_M^{App}$ determined by varying substrate A (histidine) at different fixed substrate B (Ac-CoA) concentrations, can be represented in Michaelis-Menten form by the equation $V_{max}^{App} / K_M^{App} = ((V_{max} / K_M^{His}) [Ac-CoA]) / (((K_i^{His} K_M^{Ac-CoA}) / K_M^{His}) + [Ac-CoA])$. Thus, another double-reciprocal secondary plot (Supplementary Figure 2g) of $K_M^{App} / V_{max}^{App}$ vs. $1 / [Ac-CoA]$ can be used to determine, from the Y-intercept, K_M^{His} / V_{max} , from which K_M^{His} was calculated by multiplication with V_{max} derived from Supplementary Figure 2f as described above.

We have clarified the procedure in the methods (line 692-708) and in the Supplementary figure 2 legend.

Response:

The authors answered this open point.

17. MM-kinetics: show also the primary data, i.e. [CoA] as a function of time.

RESPONSE: These data are included in the source data file.

Response:

The authors answered this open point.

18. Line 138, "...assay did not reveal the location of the acetyl group": please explain better what you want to say here, i.e. either acetylation at N-(a)- or N-(e) of lysine side chain.

RESPONSE: We have modified the text here as suggested (line 152-153).

Response:

The authors answered this open point.

19. Line 132, "...was significantly decreased": did the authors perform a statistics test? If yes show it and say in the legend what test was conducted and show the significance level.

RESPONSE: The details about the statistical tests are now included in the legend of the Figure 2 and in supplementary table 2. The statistical significance levels are indicated in Figure 2.

Response:

The authors answered this open point.

20. Fig. 2: Write "Western" with capital "W" as the name is derived from a surname.

RESPONSE: We have modified the text as suggested.

Response:

The authors answered this open point.

21. Fig. 2C: Can the authors judge how the stoichiometry of acetylation on histidine is. As shown by the metabolomics, they identified histidine and acetyl-histidine in the cells. Were both present at similar amount/concentration? Can the authors conclude anything concerning the stoichiometry?

RESPONSE: The relative abundances of the different metabolites are not comparable in this assay, so we cannot determine the relative stoichiometry of histidine vs. acetylhistidine. We have clarified this in the figure legend.

Response:

The authors answered this open point.

22. Line 165, "This agrees with a model in which": can the authors explain which model they refer to and cite the reference.

RESPONSE: The model we refer to was described in the same sentence after the word 'which', briefly, that the acetylhistidine produced by HisAT-containing cells is secreted to blood by the endocrine tissues. To clarify this, we have changed the wording of this sentence (line 188-189).

Response:

The authors answered this open point.

23. Lines 174/175: Please show a clear image of the structure from which the location of the mutation F63S becomes clear. Is it near the CoA binding site? Is it near the His binding site?

RESPONSE: We have removed the structure from Figure 3c and included a more detailed image of Phe63 in Supplementary Figure 5g. We refer to the figure and the position of the residue later when we discuss the structure (Line 259).

The residue is not very close to the active site (shortest distances to the substrate and co-substrate 14 and 15 Å, respectively), but it is located in the α 1 helix and as the α 1- α 2 loop is involved in substrate binding, it could be that mutation to serine somehow alters the orientation of this loop.

Response:

The authors answered this open point.

24. Lines 177-180: the authors show that the mutation F63S in NAT16 affects the K_M value and only slightly k_{cat} . They state that an effect is observable “especially at low histidine conditions”. It is important to note, that the concentrations matter, so replace “low histidine conditions” with “low concentrations of histidine”. If the enzyme is saturated with histidine no effect of the mutant is detectable as k_{cat} is almost unchanged. In this respect, is it known how the intracellular concentrations of histidine are in the tissues where NAT16 is expressed. Is a difference of the activity of the mutant compared to wildtype expectable? This will help to judge the physiological importance of the mutant. This could be discussed.

RESPONSE: We have modified the text as suggested.

We have found it difficult to find reliable numbers for free histidine concentration in human tissues. The range 70-120 μ M has been suggested for blood histidine concentration (Brosnan & Brosnan 2020, PMID: 33000155). Hu et al. (2017, PMID: 28252043) used a fluorescent probe to determine histidine level in glucose-fed and -starved HeLa cells in culture and found the cytosolic histidine to be 159 μ M in fed and 20 μ M in starved cells. Using the apparent K_M values for WT vs. F63S HisAT, the starved cell histidine concentration would result in roughly 3-fold slower rate for the mutant enzyme, while using the fed cell histidine concentration, the difference is 2-fold. However, we do not know how well these histidine levels match the levels in intact tissues or different types of cells, and this kind of kinetic calculation is a gross simplification of the cellular situation. We have mentioned these rate differences in the discussion (lines 505-506).

Response:

The authors answered this open point.

25. Fig. 3A,B: this is a figure derived from a public database. I suggest moving it into the SI section.

RESPONSE: As suggested, these figure panels have been moved to Supplementary Figure 4.

Response:

The authors answered this open point.

26. If seeing Fig. 3A/B the question is if the differences in the mRNA levels are manifested on the protein level in the tissues.

RESPONSE: Unfortunately, such data on tissue expression of the NAT16 protein is not available.

Furthermore, we are somewhat skeptical about the specificity of several of the commercially available NAT16 antibodies as they detect multiple bands on immunoblots but only detect NAT16 when it is overexpressed.

Response:

The authors answered this open point.

27. Fig. 3C: the structural models are not really helpful to see where the mutation is located. Is it an AlphaFold model or which structure is used here? The only difference in the two subpanels is the color. I do not see a clear benefit showing this.

RESPONSE: We agree and we have removed the structures from this panel.

Response:

The authors answered this open point.

28. Fig. 3C: the bar graphs lack correct labels on the y-axis, i.e. label and unit.

RESPONSE: We have modified the graphs to include correct labels on the Y-axis.

Response:

The authors answered this open point.

29. Check throughout if all graphs have correct labels on the axes; there are graphs in the main section and in the SI section that lack properly labelled axes.

RESPONSE: Graphs have been checked and adjusted.

Response:

The authors answered this open point.

30. Just for curiosity, can the author explain why they used isomorphous replacement to solve the phase problem and not molecular replacement? AlphaFold2 generates excellent molecular replacement models or did it not work out for the authors?

RESPONSE: The first of the NAT16 structures discussed here was solved in October 2020. At the time AlphaFold2 was not publicly available. Structure predictions at the time were not of sufficient quality to help solve the crystal structure with molecular replacement. The predictions were especially poor for the second GNAT domain that we did not identify as having a GNAT-like fold before we solved the structure.

Response:

The authors answered this open point.

31. Line 202: “coenzyme A disulfide molecule”: the authors should explain a bit better, what they want to say. i.e. it is two CoA covalently connected by formation of a disulfide bridge with their thiols.

RESPONSE: We have clarified the text (line 234-235).

Response:

The authors answered this open point.

32. Line 203: “crystallized also in space group I222”: why do the authors say “also” here; it is the first time they mention this space group in the manuscript. That is a bit confusing.

RESPONSE: This is because the shorter construct was crystallized in both space groups. We have modified the text to clarify this (line 236-237).

Response:

The authors answered this open point.

33. Line 204: “the truncated N-terminus interacted”, Line 206: “whole truncated N-terminus was visible”: these statements are really confusion. How can a part that is truncated, i.e. not present, interact with sth. or can be visible. Please work on the language and explain it differently.

RESPONSE: We have added amino acid information to improve clarity. Truncated does not mean “not present”, but “cut short” or “lacking an expected or normal element at the beginning or end” (<https://www.merriam-webster.com/dictionary/truncated>). The language refers to the N-terminus after the removal of residues 5-45 and it is correctly used here.

Response:

Thank you for clarifying this.

34. Show a 2Fo-Fc map for all ligands.

RESPONSE: We have generated a supplementary figure (supplementary figure 7) with 2Fo-Fc map for ligands.

Response:

The authors answered this open point.

35. Show a proper Fo-Fc omit map for all ligands and replace the polder map with those. A polder map is not a real omit map.

RESPONSE: We have replaced the polder maps with omit maps in supplementary figure 8.

Response:

The authors answered this open point.

36. All structure images: the images should be labelled better. Label all important parts, side chains, CoA, α -helices, β -strands, shown. Some figures are really crowded. Please work on the images to make it better readable.

RESPONSE: We have improved the clarity of all the structure figures as requested.

Response:

The authors answered this open point.

37. The ethyl-CoA is firstly mentioned in Fig. 4 and line 241 but it is explained later in lines 266 and following. Please shift this section to explain the compound when it is introduced.

RESPONSE: We have rearranged and modified the text as suggested.

Response:

The authors answered this open point.

38. Give information on how the structures with ligands were obtained, was it by soaking or by co-crystallization?

RESPONSE: The information about crystal growing and soaking solutions is included in Supplementary table 4.

Response:

The authors answered this open point.

39. Line 261: The authors state that differences in the affinities might explain the difference observed in enzyme activity between arginine and histidine and poor activity towards other amino acids. If so, the authors should observe activity under saturating concentrations of the amino acids.

RESPONSE: We have now determined the apparent kinetic parameters for HisAT also in the presence of arginine (Supplementary Figure 2h-i).

Response:

The authors answered this open point.

40. Line 257, “lower resolution cutoff”: the cutoff is set by the user, applying certain rules where to set the cutoff for the resolution, i.e. $CC1/2$ (or I/sI , R_{meas}/R_{merge}).

RESPONSE: We have replaced this text with a reference to the diffraction data statistics table.

Response:

The authors answered this open point.

41. Line 258, “The cause for the HisAT substrate preference...”. The authors explained that the imidazole ring can form quite some interactions with side chains and main chains in the enzyme’s active site. Why does not explain the specificity. Aromatic side chains, Phe and Tyr, might form the stacking interactions with the nearby aromatic side chains in the active site, and Trp might just be too bulky to fit into the active site, i.e. I would sterically not be possible to be a substrate.

RESPONSE: The only direct interactions the imidazole side chain of histidine makes with the amino acid residues in the active site arise from being stacked in between Tyr74 and Trp246. The substrate histidine interacts with the active site with the α -amino and carboxyl groups, but these interactions would presumably be shared by all amino acids (except perhaps proline), and the stacking interaction could presumably be shared at least by tyrosine and phenylalanine. However, tyrosine and phenylalanine are not efficiently acetylated by HisAT. We propose here that the acidic nature of several residues bordering the active site could contribute to the preferred substrate being histidine, but this is only our best guess at this time, which is what we attempted to convey here.

Response:

The authors answered this open point.

42. Line 271, “These two amides generate an oxyanion hole...”. Is that speculation or is that derived from structural or sequence alignments? Are these residues conserved in other GNATs? Do the authors show this in a figure?

RESPONSE: The presence of the β -bulge in most GNAT-fold enzymes is well known. The β -bulge residues forming an oxyanion hole is also well known (Vetting et al. 2005, PMID: 15581578, Salah-Uddin et al. 2016, PMID: 27367672), for example for NMT1 (Dian et al. 2020, PMID: 32111831). The residues themselves are not broadly conserved, but their orientation is. We have introduced these references here for clarity.

Response:

The authors answered this open point.

43. For the section of comparison of NAT16 with NMT1/” show a structural superposition and calculate R.M.S.D. values to get a value for their similarity.

RESPONSE: The superposition is shown in Fig. 5c and the RMSD is found in the figure legend. We have presented it in this way for all structure superpositions.

Response:

The authors answered this open point.

44. Line 304: Arg161 and Asp163 are not visible in the electron density of the substrate structure. Are they in flexible parts which are fixed upon myristoyl-histidine binding?

RESPONSE: The poor electron density concerns only the side chains, and they do still have some density. The side chain electron density appears worse when neither myristoylhistidine nor the second CoA molecule are present, suggesting they adopt more static positions in the crystal when a ligand is bound nearby. However, the difference isn't large and is only mentioned as for the most part the active sites compared here are very similar.

Response:

The authors answered this open point.

45. Line 318: “...that clearly contributes to enzymatic activity or substrate binding is Trp246”. How do the authors conclude this? Is that based on the structures showing that it is structurally involved binding the histidine? Do the authors have MM-kinetics for the mutant?

RESPONSE: This statement is based on the structural data of the substrate binding by Trp246, and its conservation among different HisAT-type proteins. We do not have enzyme kinetics data of a mutant. We have modified the statement accordingly (line 360-361).

Response:

The authors answered this open point.

46. Line 326, "... and myristoylated peptide substrate in NMT1": if the binding of substrate so similar in both enzymes, maybe this is a hint for NAT16 also being able to act as peptide/protein N-(a)-acetyltransferase if a histidine is at the N-terminus and not only active on the isolated amino acid similar to other N-terminal acetyltransferases acting co- or posttranslationally. Maybe the efficiency is higher for a peptide/protein as other parts apart from the histidine side chain contribute to binding affinity.

RESPONSE: We have corrected an error in the above quotation, the myristoylated peptide is of course a product of NMT1.

The binding of the myristoylated products is only similar for the myristoyl chain, not for the peptide or amino acid part. As seen in Figure 5 c-e; histidine binding is incompatible with it being N-terminal in a polypeptide as the polypeptide chain in the NMT1 product extends to the direction of the histidine side chain. We have now also tested NAT16 activity towards several His-starting peptides and found there to be none (Supplementary table 1).

Response:

The authors answered this open point.

47. Lines 349 and following: Can the authors explain why the neighboring genes to *nat16* 169 are so important? Do the authors have indications that these are somehow connected on the transcriptional or post-transcriptional level?

RESPONSE: The neighboring genes were used to have something to compare NAT16 to and because MOGAT3 was already reported to be missing or a pseudogene in rodents. We do not have indications that the genes are connected on a transcript level. However, it is possible that there have been chromosome level events during evolutionary history that have had an impact on multiple genes. One such event could perhaps explain why we found many fewer NAT16 and MOGAT3 genes than VGF genes in rodents.

Response:

The authors answered this open point.

48. Line 373, "poor quality": Can the authors please define how they define "poor quality" here. How can sequences be of "poor quality"?

RESPONSE: The definition was present in line 410-411. Essentially, these sequences have been "modified relative to the genome sequence to correct for possible protein-altering mismatches or indels in the genome sequence". We have changed the text to indicate the sequences were (not) labeled "low quality protein".

Response:

The authors answered this open point.

49. Lines 384 and following: the presence of NMT1/2 and NAT16 in bacteria, archaea and eukarya is a bit confusing. In line 385 the authors state that NMTs have been reported to be limited to eukaryotes while later in line 393 they state that both structures are present in archaea and bacteria. Please explain this more precisely.

RESPONSE: This comment captures quite well what we attempted to convey here. We've read that NMTs are only present in eukaryotes. We considered that perhaps NAT16-type proteins are an ancestral form of this enzyme that led to the NMT-type during eukaryogenesis. However, here we found similar structures to NMTs are also present in bacteria and archaea, making the relationship between the two groups difficult, perhaps impossible, to understand.

Response:

The authors answered this open point.

We have clarified the text to make it better understandable.

50. Might this indicate that NMT1/2 and NAT16 developed in terms of divergent evolution from a common ancestor?

RESPONSE: This is what we have attempted to understand in this section. The conclusion is that it is not clear since both types of structure are present in all domains of life suggesting either that both types were present before the split of primordial life to bacterial and archaeal lineages, or that one or both types emerged either in bacteria or archaea and were transferred to the other lineage by horizontal gene transfer. See also previous comment.

Response:

The authors answered this open point.

51. Line 443, "was not set up to find them": can the authors explain why these acylated histidines could not be detected. Can the authors re-evaluate the data?

RESPONSE: It is more accurate to state that these metabolites were not identified in the dataset, and we have amended the text in this regard. There are molecules in the dataset that have been identified by mass alone that match the molecular weight and formula of propionyl- and butyryl-histidine (identified as zalcitabine/dideoxycytidine and 4-(methylnitrosamino)-1-(3-pyridyl-n-oxide)-1-butanol, respectively). These molecules were increased in the cell lines overexpressing NAT16 but were not changed in WT vs. KO, suggesting they could be produced by overexpressed NAT16. We did not see similar patterns for molecules whose mass would match that of larger acylhistidines such as palmitoyl- or myristoylhistidine. We have mentioned the detected molecules in the discussion (line 496-498).

Response:

The authors answered this open point.

52. It is not clear how the acetyl-histidine is entering the blood? It is generated by a cytosolic enzyme. What might be the role of acetyl-histidine in cells? Can the authors speculate? Storage for free histidine? Or is it derived from peptides/proteins? Does it regulate protein function similar as described for other NATs, i.e. regulating protein stability or turnover. It could also act co-translationally if the first residue is His; or if the initiator methionine is removed, the second residue is a His. In this case it might be active towards peptides but not towards folded proteins.

RESPONSE: Based on high expression of NAT16 in endocrine cells/tissues (Supplementary figure 4), its cytosolic subcellular localization (Figure 2), and the general transport routes for amino acids across the plasma membrane, acetyl-histidine is likely to be produced in the cytosol and thereafter released into the blood. Based on our new data (Supplementary table 1) and the structural data, there are no indications that HisAT acetylates histidine in peptides or proteins.

We have discussed the potential roles of acetylhistidine, including cellular roles, in the discussion section paragraphs starting at lines 542 and 576.

Response:

The authors answered this open point.

53. Is these also non-enzymatic acetylation on histidines?

RESPONSE: We did see elevated background signal in the pH/buffer screen at pH >8.5 where a larger proportion of the histidine α -amino groups would be deprotonated potentially promoting this reaction. We are not aware of it occurring to a large extent at neutral conditions.

Response:

The authors answered this open point.

54. Do the authors have any indication if catalysis occurs by a general base mechanism. For many GNATs a catalytic Glu acts as general base. Is there a conserved Glu? Can the authors make a mutational analysis? This Glu should be in the area where the His-N-terminal amino group is located so that it can abstract a proton increasing its nucleophilicity to attack the electrophilic carbon of the acetyl group. Maybe the structures can resolve this.

RESPONSE: There are no completely conserved glutamates in the sequences shown in Supplementary Fig. 11. The mutation of the glutamate closest in the structure to the substrate histidine, Glu122 (6.7 Å from the closest O-atom to His α -N), to a glutamine slowed down the catalysis a little but did not inactivate the enzyme. Glu108 is also fairly close to the substrate (9.9 Å from the closest O-atom to

His α -N), but is not completely conserved in vertebrate sequences, and its mutation to glutamine did not reduce activity. It is possible that the deprotonation for the catalysis is mainly carried out by Glu122, but there is some redundancy in the mechanism. It is also possible that Glu122 has another role during catalysis or substrate recognition and the deprotonation is carried out by another residue, possibly transmitted via water molecules. We have included new data on relevant NAT16 mutants in Supplementary Figure 5c-d.

Response:

The authors answered this open point.

55. Methods: DpnI; write Dpn in italics; if Nat. Communications still sticks to the rule.

RESPONSE: The text was modified as suggested.

Response:

The authors answered this open point.

Reviewer #2 (Remarks to the Author): The reviewer was only tasked to review technical details of the metabolomics analyses.

The following points arise: (a) data are not uploaded to public repositories, e.g. MetaboLights or MetabolomicsWorkbench. Authors must upload both raw instrument files and result data. The authors make a false statement when they declare in the NR reporting summary that all other data are included with the manuscript. The metabolomic data has not been included in either the manuscript or the supplement. The supplement only includes compound names, RT and m/z but not MS/MS information, and not the library that was used for compound ID and MS/MS similarity thresholds. The supplement does not include the actual data, but only the statistics results. The supplement does not include the instrument raw data, either.

RESPONSE: The data are now uploaded to MetaboLights database with the id MTBLS12070 and are accessible via the following URL (<https://www.ebi.ac.uk/metabolights/reviewer6f53c866-cd81-4513-a38d-a14a46266555>).

(b) cells were extracted by methanol/chloroform (80:20), excluding highly hydrophilic compounds such as sugar acids. This is probably fine with the aim of the analyses that looked at amino acids, but should be discussed as limitation. Given the very lipophilic extraction method, it is strange to see that the authors did not detect lipids in their 'semi-polar' chromatography method.

RESPONSE: We have included a sentence about the extraction method excluding highly hydrophilic compound in the results section. There were some lipids detected, but these were only identified based on mass (level 3).

(c) The authors do not disclose the chromatography methods, but refer to two former publications, and even say that methods were modified ("using a slightly modified version of the protocol described by Hsiao et al.¹³² for polar metabolites or by Doneanu et al. for semi-polar metabolites."). This is not acceptable reporting practice. Methods must be given in detail to allow other researchers to repeat analyses. Method details can also be given in supplements, if needed, and if method sections get too long otherwise.

RESPONSE: The detailed chromatography methods are included in the MetaboLights submission and in the methods section including the supplementary table 10.

(d) Compound annotations is likely correct, and authors give some criteria (but not RT windows and not MS/MS similarity thresholds). Software Compound Discoverer is fine, but the authors do not report the confidence levels reported by that software, and authors do not report the parameters used for data processing. A number of compounds are not fully annotated ('hexose' or 'sugar alcohol'), and it is unclear which external library was used for compound annotations.

RESPONSE: These details are now included in the MetaboLights submission and the methods section. Accepted RT deviation was 0.2 min, accepted mass deviation 3 ppm, and MS/MS match factor was >60.

Reviewer #3 (Remarks to the Author):

The paper by Myllykoski et al provides new, important information on Nat16, as the enzyme that acetylates histidine in specific cells and tissues in humans. Until now, this function had only been described in a fish, and the human recombinant enzyme had been tested, but was apparently without any activity. The function is now clear and the role of Nat16 needs to be found. Nat16 is present in many vertebrates and in more primitive organisms, but it is apparently absent from most rodents, though, as stressed by the authors, the difficulty of sequencing Nat16 in many species, makes that it is difficult to be sure about this absence. The paper provides a quite complete kinetic investigation of Nat16 and provides its structure, with its peculiar double GNAT fold, which is compared with that of other enzymes of the GNAT family.

Overall the paper provides a lot information and provides a good coverage of the literature. This paper is a major step in the elucidation of the function of N-acetylhistidine, which is predominantly expressed in brain and secretory tissues. The experiments have been carefully performed and well analysed.

I have only minor comments

Summary Line 22 ...Lower blood levels of acetylhistidine which impacts kidney function
This a strong statement: nothing is known on how a reduced acetylhistidine level could cause kidney dysfunction.

RESPONSE: We have removed this strong statement.

Figure 1. Arginine and lysine are substrates for Nat16, as shown in Fig1A. Since the results in Fig. 1a underestimate the activity on histidine, it would be good to provide values of K_m and V_{max} of the enzyme when lysine or arginine are used as a substrate. There is no need to perform as detailed a kinetic investigation as performed for histidine.

RESPONSE: We have carried out enzyme assays with varied arginine and lysine concentrations at fixed Ac-CoA concentration. The results are included in Supplementary Figure 2h-i and indicate that both are on a similar level as substrates whereas the catalytic rate is slightly lower and K_M slightly higher for lysine. Both have more than one order of magnitude difference to histidine in both k_{cat} and K_M . Together with the metabolomics data, these results suggest that arginine and lysine are probably not substrates for NAT16 except in exceptional situations, like when NAT16 is artificially overexpressed.

Line 117. Please provide information on the concentration of histidine used for determining the kinetic constant in Fig. 1C

RESPONSE: We have elaborated on details of the enzyme assays in the Figure 1 legend.

Figure 2C : overexpression of Nat16 seems to cause higher increases in acetyllysine than in N-acetylarginine, yet arginine seems to be a better substrate on the purified enzyme than lysine. What does 'relative abundance' mean ? Are the values for the different metabolites comparable ?

RESPONSE: The relative abundance is derived from the area under the curve of the chromatography peak. These values are not comparable between different metabolites, so we cannot use these results to compare which is better *in vivo* substrate, arginine or lysine, or to determine how much of the cellular histidine pool is acetylated. We have clarified this in the figure legend.

Line 609, page 25; were the kinetic studies performed with a whole set of Acetyl-CoA concentrations combined with a whole set of histidine concentration, as the sentence suggests ?

RESPONSE: Yes. For the first two series we used slightly staggered setup where the histidine concentrations for the three lowest Ac-CoA concentrations were slightly different than for the highest three (0, 20, 50, 120, 300, 720, 1800, or 4500 μM histidine for lower Ac-CoA concentrations in Series 1 and 2 and all Ac-CoA concentrations in Series 3, and 0, 50, 120, 300, 720, 1800, 4500, or 10000 μM histidine for higher Ac-CoA concentrations in Series 1 and 2). We have clarified these in the method section.

Supplementary figure 2, legend; please indicate the concentration of histidine that was used

RESPONSE: We have included the concentrations in the legend.